# What Makes Good Examples for
# Visual In-Context Learning?

**Yuanhan Zhang**[1]    **Kaiyang Zhou**[2]    **Ziwei Liu**[1, ✉]
[1]S-Lab, Nanyang Technological University, Singapore
[2]Hong Kong Baptist University, Hong Kong
{yuanhan002, ziwei.liu}@ntu.edu.sg
kyzhou@hkbu.edu.hk

## Abstract

Large vision models with billions of parameters and trained on broad data have great potential in numerous downstream applications. However, these models are typically difficult to adapt due to their large parameter size and sometimes lack of accesss to their weights—entities able to develop large vision models often provide APIs only. In this paper, we study how to better utilize large vision models through the lens of in-context learning, a concept that has been well-known in natural language processing but has only been studied very recently in computer vision. In-context learning refers to the ability to perform inference on tasks never seen during training by simply conditioning on in-context examples (i.e., input-output pairs) without updating any internal model parameters. To demystify in-context learning in computer vision, we conduct an extensive research and identify a critical problem: downstream performance is highly sensitivie to the choice of visual in-context examples. To address this problem, we propose a prompt retrieval framework specifically for large vision models, allowing the selection of in-context examples to be fully automated. Concretely, we provide two implementations: (i) an unsupervised prompt retrieval method based on nearest example search using an off-the-shelf model, and (ii) a supervised prompt retrieval method, which trains a neural network to choose examples that directly maximize in-context learning performance. Both methods do not require access to the internal weights of large vision models. Our results demonstrate that our methods can bring non-trivial improvements to visual in-context learning in comparison to the commonly-used random selection.

## 1   Introduction

In recent years, large-scale models have emerged in computer vision: they have enormous parameter sizes and are pre-trained on broad data to gain wide-ranging knowledge. These models have demonstrated remarkable generalization performance and have great potential for numerous downstream applications [4]. However, due to the large model size and the potentially proprietary data used for training, entities able to develop large-scale models typically only provide users with APIs, known as Model-as-a-Service (Maas). Representative examples include the prominent text-to-image generation models, DALL·E [17] and Imagen [20], and OpenAI's powerful language models like GPT-3/ChatGPT [16]. As a result, users are unable to apply full fine-tuning or some parameter-efficient tuning techniques, such as prompt learning [11, 10, 26, 25, 23, 15], for model adaptation, largely limiting downstream performance. To circumvent the problem, "black-box" adaptation methods need to be developed.

---

[✉]Corresponding author

37th Conference on Neural Information Processing Systems (NeurIPS 2023).

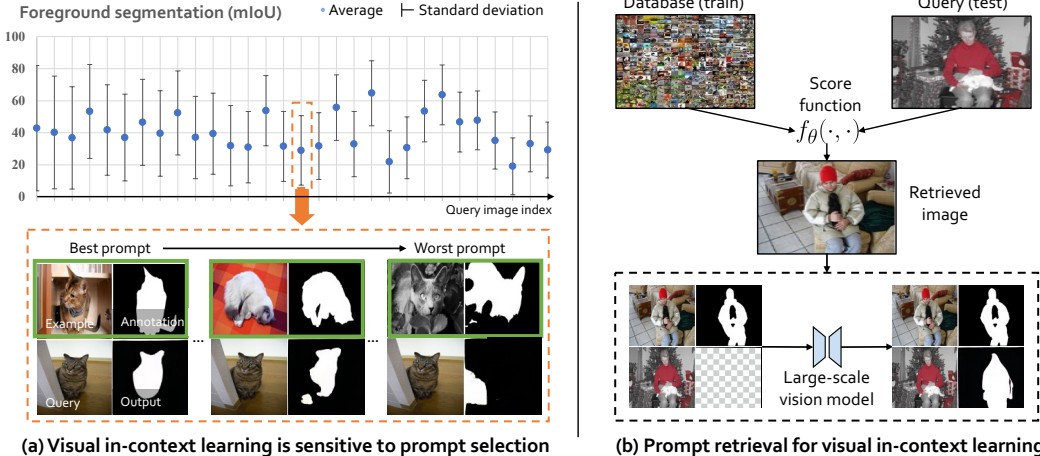

**(a) Visual in-context learning is sensitive to prompt selection**    **(b) Prompt retrieval for visual in-context learning**

Figure 1: (a) Different choices of in-context examples (outlined in green) often lead to significantly different results. Here we show 30 random query images (x-axis) from Pascal-$5^i$ [21] split 0, and measure the performance range using 50 different in-context examples. (b) We propose a prompt retrieval framework aiming to automate the selection of in-context examples. We provide two implementations of the idea: one is unsupervised while the other is supervised, both outperforming random selection by a clear margin and without accessing the large vision model's internal weights.

*In-context learning*, which is a "hidden" capability originally found in large autoregressive language models [16], has recently been investigated for large vision models [3], and more importantly, has the potential to be adopted as the mainstream approach in MaaS applications in the future. Without the need to update any parameter when deploying in previously unseen tasks, in-context learning simply prepends some domain-specific input-output pairs, called in-context examples or prompt,[2] to a test example, which together guide the model to produce an ideal result. For instance, in natural language processing one could prepend a French-English sentence pair to a French sentence, and the model would produce an English translation of the French sentence. In computer vision, Bar *et al.*. [3] pre-trained a neural network to fill missing patches in grid-like images, which allows the model to perform in-context learning for unseen tasks like image segmentation (see the grid images in Fig. 1(a) bottom).

In this work, we focus on *visual in-context learning*, a relatively new concept with little existing research regarding how to better apply it in practice. We for the first time conduct a comprehensive investigation on the impact of in-context examples for large vision models, and identify a critical issue: downstream performance is highly sensitive to the choice of in-context examples. This is evidenced by the large variances observed for a variety of test examples shown in Fig. 1(a) top. By visualizing the results in Fig. 1(a) bottom, it seems to suggest that the closer the in-context example to the query, the better the result. For example, the best prompt image is closer to the query as they are similar in object pose and background; on the other hand, the worst prompt image has a drastically different style than the query image, which might explain why the predicted mask focuses on the wrong region, i.e., the white pillar instead of the cat.

Clearly, designing a proper prompt containing the optimal in-context example(s) by hand would be extremely difficult. To overcome the problem, we propose a prompt retrieval framework where the core component is a score function, which aims to give each source instance a score to indicate the level of suitability for being included in the prompt. Once the scoring process is done, we can simply pick one or multiple examples with the highest score(s) to construct a prompt. An overview of our framework is depicted in Fig. 1(b).

We provide two implementations for the prompt retrieval framework, both interpreting the score as the cosine distance measuring similarity between a query and a source example. The first is an unsupervised method based on nearest example search using an off-the-shelf model. The second

---

[2]To clarify, in natural language processing, a prompt refers to the instruction input specifying a particular purpose while in-context examples are a special prompt, which consist of domain-specific input-output pairs.

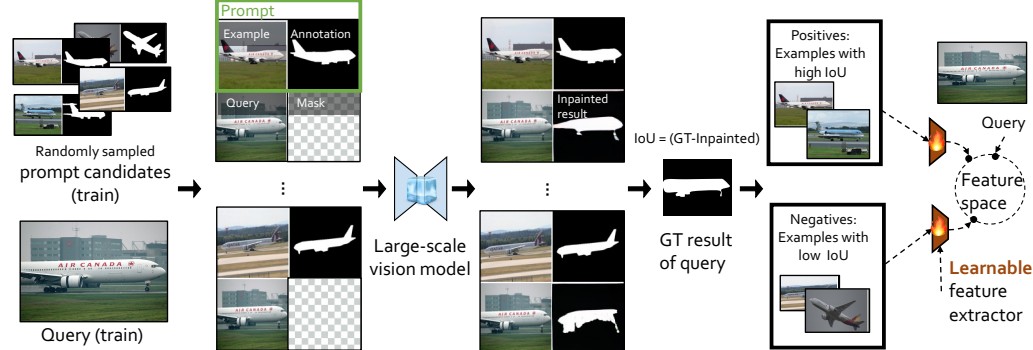

Figure 2: Overview of the supervised prompt retrieval method. The main idea is to compute the in-context learning result for each source example, and pick those with the highest/lowest results to form a positive/negative set for contrastive learning.

is a supervised method, which learns a neural network to choose examples that directly maximize in-context learning performance. Since there is no ground-truth score to be used as the supervisory signal, we resort a contrastive learning paradigm: source examples that result in better (or worse) in-context learning performance should get closer (or farther) to the query in feature space.

Our contributions and the main findings are summarized as follows. (1) We present the first comprehensive study concerning how to select good examples for the emerging visual in-context learning, and reveal a critical issue that the choice of in-context examples has a huge impact on performance. (2) From the technical perspective, we present a prompt retrieval framework that can automate the prompt selection process, and provide two simple implementations: an unsupervised method and a supervised method. (3) By conducting extensive experiments on three visual in-context learning tasks (which have not been seen during pre-training), namely foreground segmentation, single object detection and image colorization, we share valuable insights with the community on how to find good visual in-context examples, e.g., the supervised method performs the best and often finds examples that are both semantically close and spatially similar to a query. (4) The code and models are available at https://github.com/ZhangYuanhan-AI/visual_prompt_retrieval.

## 2   Methods

### 2.1   Visual In-Context Learning

In-context learning is a new paradigm that originally emerged from large autoregressive language models pre-trained on broad data, such as GPT-3 [5]. Unlike traditional learning methods, in-context learning does not require any parameter update and instead conditions prediction on some in-context examples in the form of input-output pairs. For example, in natural language processing one might give a French-English sentence pair and a test French sentence as input to the model, which then produces the English version of the sentence. In computer vision, such a paradigm has only been studied very recently. For example, Bar *et al.* [3] trained a neural network to fill missing patches in grid-like images, which in turn allows the model to perform in-context learning on unseen tasks.

Formally, given a dataset $\mathcal{D} = \{(x_n, y_n)\}_{n=1}^{N}$ containing $N$ image-label pairs (e.g., an image and its segmentation mask), a query example $x_q$, and a model $g_\tau$, in-context learning can be formulated as:

$$y_q = g_\tau(\mathcal{P}, x_q), \tag{1}$$

where $\mathcal{P}$ is called a prompt, which consists of $K$ input-output pairs, $\mathcal{P} = \{x_{c_1}, y_{c_1}, ..., x_{c_K}, y_{c_K}\} \subset \mathcal{D}$. In particular, the prompt $\mathcal{P}$ provides some *context* for guiding the model to produce the ideal $y_q$ for $x_q$ without updating the large model's parameters $\tau$.

**Problem.** The most common approach for designing the prompt $\mathcal{P}$ in the vision domain is (within-class) *random selection* proposed by [3]: one or multiple image-label pairs (with the same label as the test example) are randomly chosen from the training dataset. As illustrated in Fig. 1(a), the

performance is highly sensitive to the selection of in-context examples—the gap between the best and worst prompt could reach over 70% mIoU. Below we propose two automatic prompt selection methods to tackle this problem.

## 2.2 Prompt Retrieval

Our goal is to automatically select the most suitable example(s) from the training dataset for a query $x_q$. To this end, we propose a prompt retrieval framework in the following form,

$$x^* = \arg \max_{x_n \in \mathcal{D}} f_\theta(x_n, x_q), \tag{2}$$

where $f_\theta$ is a function parameterized by $\theta$, aiming to produce a score for a pair of $x_n$ and $x_q$. When $K = 1$, we choose the optimal example pair as the prompt, $\mathcal{P} = \{x^*, y^*\}$. When $K > 1$, we rank the training examples by their scores and choose the top-$K$ example pairs. An overview of our methods is provided in Fig. 1(b).

In this work, we implement $f_\theta$ as a combination of a neural network for feature extraction and the cosine distance function for measuring similarity between two feature vectors.

### 2.2.1 Unsupervised Prompt Retrieval

Our first method is *unsupervised prompt retrieval* where the key idea is to use an off-the-shelf feature extractor for extracting image features so that we can compare the cosine distance between the query $x_q$ and each training example $x_n \in \mathcal{D}$. In this case, the parameters $\theta$ for the score function $f_\theta$ correspond to the off-the-shelf feature extractor, which are kept fixed.

### 2.2.2 Supervised Prompt Retrieval

The unsupervised method discussed above is not explicitly optimized for in-context learning; instead, it relies on how the feature extractor was pre-trained and the objective (function) used in pre-training may well not align with that of in-context learning. We propose a second method based on *supervised prompt retrieval* where we assume the source data contains labels. The goal is to directly optimize the score function $f_\theta$ such that the chosen in-context example(s) can maximize the log-likelihood,

$$\max_{\mathcal{P}} \quad \log p(y_q | \mathcal{P}, x_q). \tag{3}$$

In this work, we present a simple implementation for the supervised method, which simply turns the unsupervised method into a supervised one by making the feature extractor learnable. In other words, we directly optimize Eq. 3 with respect to the feature extractor. Below we explain in detail how we train the feature extractor (see Fig. 2 for an overview).

**Data.** Recall that we interpret the score $f_\theta(\cdot, \cdot)$ as the cosine distance between two images in feature space. We would like to learn a space such that an image pair $(x_n, x_q)$ with high in-context learning performance is close to each other, or far away from each other if the performance is low. Since there is no label defining how close a distance should be, we resort to contrastive learning for training the feature extractor. The goal is then to find a positive and a negative set for each training example $x_n \in \mathcal{D}$ treated as a query. Specifically, for each example $x_n$ we compute the prediction $\hat{y}_n = g_\tau((x_m, y_m), x_n)$ where $g_\tau$ is the large vision model defined in Sec. 2.1 and $x_m \in \mathcal{D}$ but $x_m \neq x_n$. Since we have the ground truth $y_n$ for $x_n$, we can measure the performance by comparing the prediction $\hat{y}_n$ with the ground truth $y_n$. Then, for each $x_n$ we choose the top-5 examples with the highest/lowest performance to form a positive/negative set.

**Training.** Let $z_n$ denote the features of $x_n$ extracted by the neural network we aim to optimize. At each iteration, we sample a mini-batch $\mathcal{B}$ from the training dataset. Then, for each example in $\mathcal{B}$, we sample one example from the top-5 positive and negative sets, respectively. The contrastive loss is computed as

$$\ell = -\frac{1}{|\mathcal{B}|} \sum_{x_n \sim \mathcal{B}} \log \frac{e^{cos(z_n, z_n^+)}}{e^{cos(z_n, z_n^+)} + \sum_{z_n^- \in \mathcal{N}} e^{cos(z_n, z_n^-)}}, \tag{4}$$

Table 1: Main results. The two prompt retrieval methods outperform random selection, and the supervised method achieves the best performance.

| | Seg. (mIoU) ↑ | | | | | Det. (mIoU) ↑ | Color. (mse) ↓ |
|---|---|---|---|---|---|---|---|
| | Split-0 | Split-1 | Split-2 | Split-3 | Avg | | |
| Random | 28.66 | 30.21 | 27.81 | 23.55 | 27.56 | 25.45 | 0.67 |
| UnsupPR | 34.75 | 35.92 | 32.41 | 31.16 | 33.56 | 26.84 | **0.63** |
| SupPR | **37.08** | **38.43** | **34.40** | **32.32** | **35.56** | **28.22** | 0.63 |

where $cos(\cdot, \cdot)$ is the cosine distance function, $z_n^+$ denotes the feature representation of a positive example, and $z_n^-$ denotes the feature representation of a negative example. It is worth noting that for mini-batch training, the negative set $\mathcal{N}$ contains a negative example of $x_n$ sampled from the top-5 negative set and other examples within the same mini-batch.

# 3 Experiments

In this section we conduct a comprehensive evaluation using different prompt selection methods (Sec. 3.1) and compare their robustness to distribution shifts (Sec. 3.2). We also provide extensive quantitative and qualitative analyses in Sec. 3.3 to help understand why our methods work and how to better apply them in practice. Source code will be released to the community for reproducing the full experiments.

**Methods.** All experiments are based on the image inpainting model pre-trained by [3] on a dataset consisting of academic figures.[3] This inpainting model comprises an encoder and a decoder, as depicted in Figure 2. The encoder generates a feature vector as its output, while the decoder yields a reconstructed image of the masked part of the input image. We mainly compare the following methods: (1) *Random*, the baseline method that randomly samples in-context examples from the source training dataset; (2) *Unsupervised prompt retrieval (UnsupPR)*, our first proposed method that uses off-the-shelf features for nearest example search. The main experiments are based on CLIP's vision encoder [16], which was pre-trained using multimodal contrastive learning; (3) *Supervised prompt retrieval (SupPR)*, our second proposed method that fine-tunes CLIP's vision encoder by directly optimizing in-context learning performance on downstream datasets. A variety of backbones are evaluated in Sec. 3.3.

**Training details for the supervised model.** The supervised model is trained for 200 epochs using SGD. The initial learning rate is set to 0.005, decayed by the cosine annealing rule. Each task need a specific supervised model for SupPR.

## 3.1 Main Results

**Setup.** Following [3], we evaluate our methods on three computer vision tasks, which have not been seen during the training of the image inpainting model. We provide the details about the datasets used for these tasks as follows. (1) *Foreground segmentation*: We use Pascal-$5^i$ [21], which has four non-overlapping splits each containing five categories. The results are averaged over all splits. (2) *Single object detection*: The experiments are done on Pascal VOC [7]. (3) *Colorization*: We use ImageNet-2012 [19], where the original validation set containing 50,000 images is used as our test set. The training data used to learn our supervised prompt retrieval model is created by randomly sampling 50,000 images from ImageNet's 1.2M training set. For all experiments, in-context examples come from the training set.

**Results.** Table 1 shows the results on the three benchmarks covering foreground segmentation, single object detection, and colorization. We summarize our findings as follows. *First, prompt retrieval clearly outperforms random selection*. In particular, the improvements of prompt retrieval over random selection are significant in foreground segmentation and single object detection: more than 6% on the former and 1% on the latter. However, the gains on colorization are only marginal (0.63 vs. 0.67), suggesting that the image inpainting model is probably weak at image coloriza-

---

[3]https://github.com/amirbar/visual_prompting

tion. *Second, the supervised prompt retrieval method performs the best.* This is not surprising as the supervised method optimizes in-context learning performance concerning the prompt selection module. In contrast, the unsupervised method relies more on the off-the-shelf feature extractor. Overall, the results well justify the design of the prompt retrieval framework, which can serve as a strong baseline for future research.

## 3.2 Experiments on Distribution Shifts

**Setup.** Distribution shifts are commonly seen in real-world applications, and therefore AI models need to be robust to distribution shifts [24]. To test this ability in visual in-context learning, we create a new protocol focusing on foreground segmentation where the source dataset is Pascal while the target dataset is MSCOCO [12]. Specifically, we follow the design of Pascal-$5^i$ and create MSCOCO-$5^i$, which also has four splits, each having the same set of categories as in the corresponding split in Pascal-$5^i$. Note that such a shift mainly affects the supervised prompt retrieval method that requires training but not the unsupervised UnsupPR and Random.

**Results.** The results are shown in Table 2. First of all, the unsupervised prompt retrieval method beats the random selection method by a clear margin. By comparing the two prompt retrieval methods, we find that the supervised method again performs better than the unsupervised one despite being a learning-based approach—this is an exciting finding as it means the supervised method does not have the overfitting problem here. Nonetheless, we observe

Table 2: Results on distribution shifts (from Pascal to MSCOCO). Despite being a learning-based approach, SupPR shows stronger robustness than UnsupPR and Random, which do not require any training.

|  | Seg. (mIoU) ↑ | | | | |
|---|---|---|---|---|---|
|  | Split-0 | Split-1 | Split-2 | Split-3 | Avg |
| Random | 12.17 | 18.47 | 20.55 | 15.94 | 16.78 |
| UnsupPR | 12.67 | 19.62 | 21.33 | 18.44 | 18.02 |
| SupPR | **13.62** | **21.25** | **24.46** | **20.44** | **19.95** |

that the gains achieved by the prompt retrieval methods here are generally smaller than the gains achieved on the standard foreground segmentation benchmark: here SupPR is only around 3% better on average than Random (19.95% vs.16.78%) while the improvement in Table 1 reaches 8% (35.56% vs. 27.56%). One potential solution to reduce the gap might be to improve the image inpainting model, which is beyond the scope of this paper.

## 3.3 Further Analysis

**What are good in-context examples?** To answer this question, we visualize the in-context examples found by UnsupPR and SupPR in Fig. 3. We focus on foreground segmentation and choose two categories from Pascal (person and cow). In each grid, the first row corresponds to the retrieved in-context example (i.e., an input-output pair) while the second row contains the query and model prediction. By comparing the in-context examples picked by UnsupPR and those picked by SupPR, we find the reason why SupPR performs better than UnsupPR: the examples found by SupPR are more similar to the queries in terms of semantics (e.g., Fig. 3(e)), background (e.g., Fig. 3(a)), object pose (e.g., Fig. 3(b), object appearance (e.g., Fig. 3(i)), viewpoint (e.g., Fig. 3(k)), and so on. We also observe similar patterns in other categories/tasks (please refer to the supplementary).

**Backbone.** To understand if using a different backbone than CLIP would make a big difference, we further evaluate our prompt retrieval methods, UnsupPR and SupPR, on the foreground segmentation benchmark using two other backbones: EVA [8] pre-trained using self-supervised learning (i.e., masked image modeling) and ViT [6] pre-trained using supervised learning. The results are reported in Table 3. Although these three backbones perform differently on image recognition under the fine-tuning setting—EVA performed the best—the gap between them

Table 3: Comparison between different backbones pre-trained using different methods: multimodal contrastive learning for CLIP, self-supervised learning for EVA, and supervised learning for ViT. Overall, the performance is insensitive to the choice of backbones.

|  |  | Seg. (mIoU) ↑ | | | | |
|---|---|---|---|---|---|---|
|  |  | Split-0 | Split-1 | Split-2 | Split-3 | Avg |
| UnsupPR | CLIP | 34.75 | 35.92 | **32.41** | 31.16 | 33.56 |
|  | EVA | 34.75 | 36.09 | 32.11 | **31.61** | 33.64 |
|  | ViT | **35.10** | **37.37** | 32.05 | 30.80 | **33.83** |
| SupPR | CLIP | **37.08** | 38.43 | 34.40 | 32.32 | 35.56 |
|  | EVA | 36.11 | 39.14 | 34.31 | **33.30** | 35.71 |
|  | ViT | 36.80 | **39.70** | **34.71** | 33.25 | **36.12** |

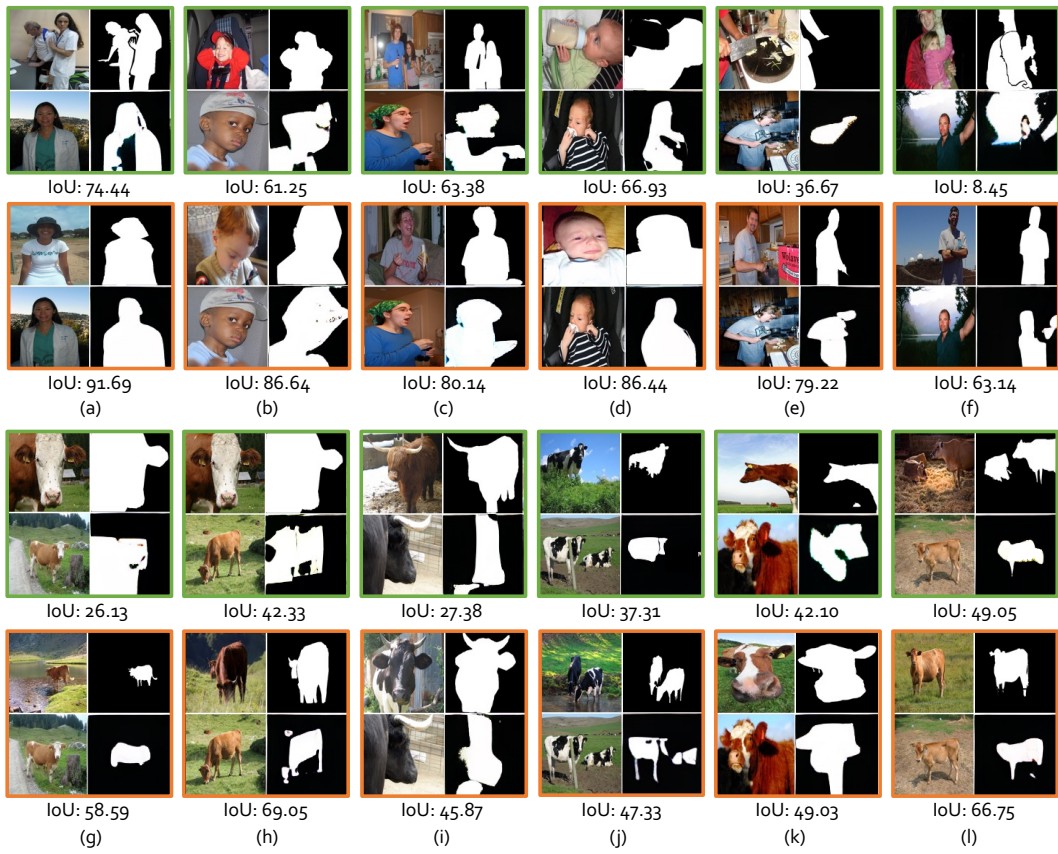

| IoU: 74.44 | IoU: 61.25 | IoU: 63.38 | IoU: 66.93 | IoU: 36.67 | IoU: 8.45 |

| IoU: 91.69 | IoU: 86.64 | IoU: 80.14 | IoU: 86.44 | IoU: 79.22 | IoU: 63.14 |
| (a) | (b) | (c) | (d) | (e) | (f) |

| IoU: 26.13 | IoU: 42.33 | IoU: 27.38 | IoU: 37.31 | IoU: 42.10 | IoU: 49.05 |

| IoU: 58.59 | IoU: 69.05 | IoU: 45.87 | IoU: 47.33 | IoU: 49.03 | IoU: 66.75 |
| (g) | (h) | (i) | (j) | (k) | (l) |

Figure 3: In-context examples retrieved by UnsupPR and SupPR. In each grid, the first row contains the prompt while the second row contains the query and prediction. The in-context examples found by SupPR are more similar than those found by UnsupPR to the queries in a numer of ways: semantics (e.g., (e)), background (e.g., (a)), object pose (e.g., (b), object appearance (e.g., (i)), viewpoint (e.g., (k)), etc. More examples can be found in the supplementary.

Table 4: Impact of the order of in-context examples. The results of different orders are averaged and their standard deviations are computed. The small variations in the results suggest that the order of in-context examples does not matter.

|  | **Seg. (mIoU) ↑** | | | | |
|  | Split-0 | Split-1 | Split-2 | Split-3 | Avg |
|---|---|---|---|---|---|
| Random | $17.93 \pm 0.20$ | $25.48 \pm 0.27$ | $21.34 \pm 0.73$ | $21.12 \pm 0.53$ | $21.46 \pm 0.43$ |
| UnsupPR | $20.22 \pm 0.31$ | $27.58 \pm 0.40$ | $22.42 \pm 0.38$ | $23.36 \pm 0.42$ | $23.39 \pm 0.37$ |
| SupPR | $\mathbf{20.74} \pm 0.40$ | $\mathbf{28.19} \pm 0.37$ | $\mathbf{23.09} \pm 0.34$ | $\mathbf{24.22} \pm 0.48$ | $\mathbf{24.06} \pm 0.40$ |

for both UnsupPR and SupPR is less than 1%. Therefore, we can conclude that the backbone for visual in-context learning does not matter much.

**Number of in-context examples.** We follow [3] and create a large grid enough to fit 8 examples at maximum (as shown in Fig. 4 right). By varying the number of in-context examples from 1 to 7, we obtain a set of results and plot them in Fig. 4 left. Clearly, more in-context examples lead to better performance for all three methods, including SupPR, UnsupPR, and Random. This is probably because in-context examples can be viewed as "training data", and having more training data typically benefits performance—in visual in-context learning, more training data gives a more comprehensive "context." We show a few example cases in Fig. 4 right to explain this observation.

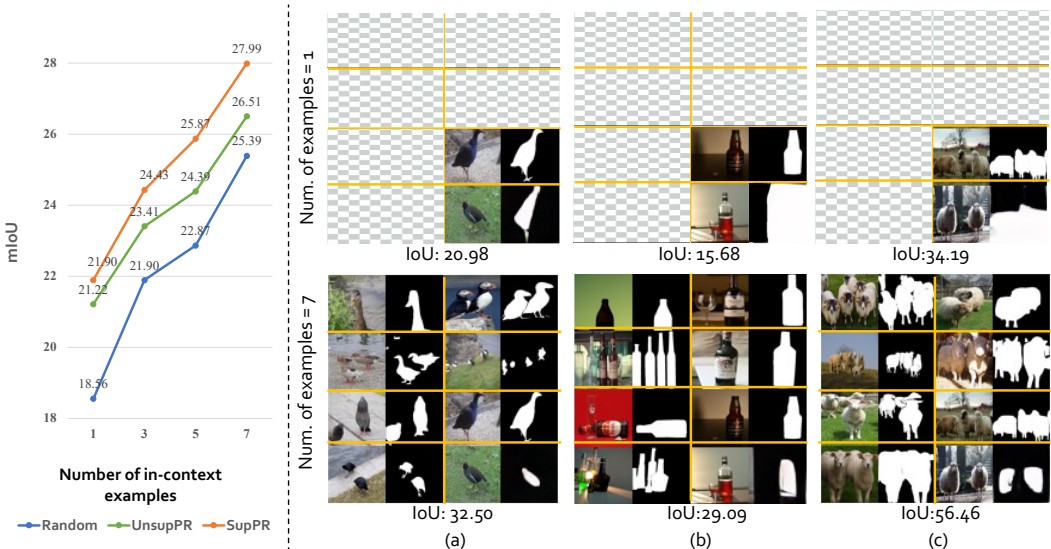

Figure 4: (Left) Impact of the number of in-context examples. (Right) More in-context examples can lead to better performance. The query in each grid is shown in the bottom right.

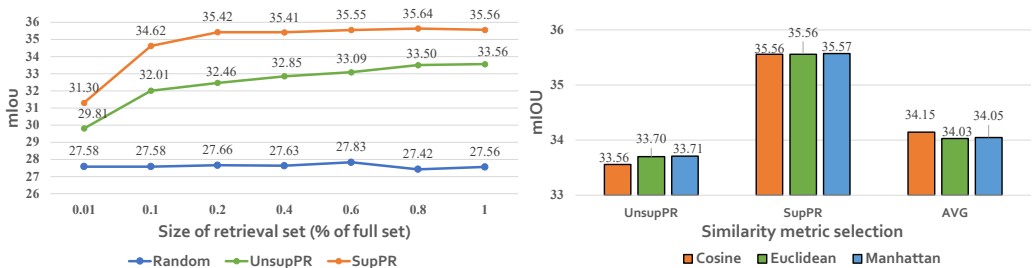

Figure 5: (Left) Impact of the size of retrieval set. (Right) Ablation study on distance metric used to compute the score function in Eq. 2. It can be observed that different metrics perform similarly.

**Order of in-context examples.** To understand if changing the order of in-context examples makes a difference, we fix the number of in-context examples to 3, evaluate all possible combinations, and compute the mean and standard deviation. As shown in Table 4, the standard deviation is generally small, so the order is not a concern as long as good examples are chosen.

**Size of retrieval set.** Recall that in-context examples are sampled from the training dataset, namely the retrieval set. We are interested to know whether the size has any impact on performance, especially for the supervised prompt retrieval method. To this end, we build seven subsets for each split in Pascal-$5^i$, which cover a wide range of sizes (see the x-axis in Fig. 5 left). The results are plotted in Fig. 5 left. For random selection, the size does not matter at all. In contrast, the two prompt retrieval methods clearly benefit from a bigger size. But their performance plateaus when the size reaches a certain level. It is worth noting that for the supervised method, 20% of the total data is sufficient for achieving a decent performance.

**SupPR vs. a supervised learning model.** To justify that the strength of SupPR comes from its design rather than the supervision signal, we further train a supervised learning model for each task and use the trained model to extract image features for prompt selection. This can be seen as replacing the off-the-shelf model in Un-

Table 5: Comparison between SupPR and a fine-tuned MAE.

|  | Seg. (mIoU) ↑ | | | | |
|---|---|---|---|---|---|
|  | Split-0 | Split-1 | Split-2 | Split-3 | Avg |
| MAE | 31.69 | 34.26 | 30.01 | 28.55 | 31.27 |
| SupPR | **37.08** | **38.43** | **34.40** | **32.32** | **35.56** |

supPR with a supervised learning model. In this experiment, we fine-tune the state-of-the-art Masked Autoencoder (MAE) model [9] on the Pascal-$5^i$ dataset, and use the image encoder for feature extraction. Table 5 shows the results where SupPR largely outperforms the supervised MAE model.

**Distance metric.** We use the cosine distance by default to compute the score function in Eq. 2. Here we evaluate other design choices including Euclidean distance and Manhattan distance. As shown in Fig. 5 right, the results are very similar for different distance metrics.

## 4 Related Work

### 4.1 In-Context Learning

In-context learning is a novel learning-free paradigm that originally emerged in large language models, such as GPT-3 [5]. In computer vision, in-context learning is a relatively new concept. One of the earliest studies tackling in-context learning is Flamingo [2], a large visual language model that can be prompted with textual instructions and can cope with both images and video sequences. More relevant to our work is an image inpainting model developed by Bar *et al.* [3], which is pre-trained to fill missing patches in academic paper figures and capable of performing in-context learning on new tasks like foreground segmentation and image colorization. Another related work is Painter [22], a generalist model that is trained on the combination of a variety of computer vision tasks (e.g., depth estimation and semantic segmentation) using customized output spaces. Our work mainly follows the image inpainting model [3] where the pre-training task differs significantly from the downstream task, and reveals valuable insights about visual in-context learning.

### 4.2 Prompt Retrieval in NLP

The natural language processing community has found that the choice of in-context examples has a huge impact on performance [1, 13]. Moreover, the way how in-context examples are constructed can also affect performance, e.g., prompt length and order of in-context examples. These findings prompted the NLP community to study how to find good in-context examples for large language models, which largely inspired our research. Liu *et al.* [13] assumed that good in-context examples should be semantically close to query sentences, based on which they proposed to select nearest neighbors in the training set measured by a sentence encoder like RoBERTa [14]. Rubin *et al.* [18] first used an unsupervised method to retrieve some candidates, among which top examples were chosen using a supervised prompt retriever to maximize downstream performance. Different from the prompt retrieval research in NLP, our work focuses on large vision models, which are different from language models in many ways and require customized designs to deal with visual input.

## 5 Discussion and Conclusion

Our research presents a timely study on an emergent ability termed in-context learning for large vision models. We systematically investigate how the choice of in-context examples impacts downstream performance, exposing a critical issue that different in-context examples could lead to drastically different results. We then propose an effective prompt retrieval framework for visual in-context learning, with two simple implementations provided: one based on unsupervised learning and the other based on supervised learning. Our methods obtain significant improvements over random selection under various problem settings, demonstrating huge potential of prompt retrieval methods in black-box adaptation scenarios for vision foundation models.

Our research also unveils some intriguing phenomena. For instance, we show that a good in-context example should be semantically similar to the query and closer in context. A model that can better balance spatial and semantic closeness in feature space would be more ideal for visual in-context learning. We hope the insights presented in this work could pave the way for developing more effective prompt retrieval methods.

Our experiments show that our methods are not strong enough to cope with distribution shifts. Though our methods outperform random selection under distribution shifts, the gap is much smaller than that on a standard benchmark, suggesting huge room for improvement.

## Acknowledgement

This study is supported by the Ministry of Education, Singapore, under its MOE AcRF Tier 2 (MOE-T2EP20221-0012), NTU NAP, and under the RIE2020 Industry Alignment Fund – Industry Collaboration Projects (IAF-ICP) Funding Initiative, as well as cash and in-kind contribution from the industry partner(s).

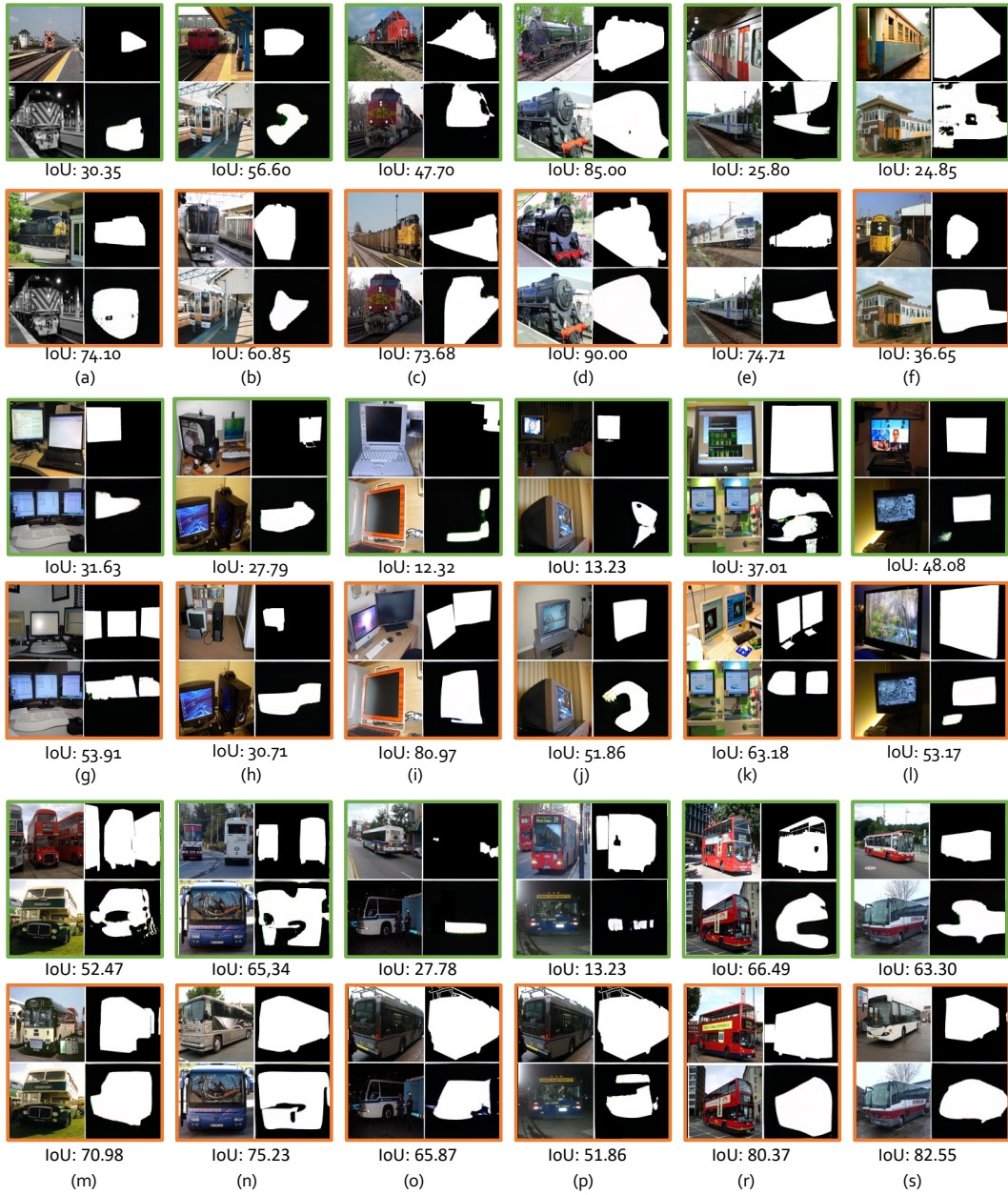

Figure 6: In-context examples, which are from the foreground segmentation task, retrieved by UnsupPR and SupPR. These grids show examples from the train, tv, and bus categories.

## A    Illustration of In-context Examples

In the supplementary material, we illustrate more in-context learning results of foreground segmentation, single object detection, and colorization tasks.

### A.1    Foreground Segmentation

The main paper presents the in-context examples from the person and cow categories. In the supplementary, as shown in Fig. 6-11, we present examples from the remained 18 categories in Pascal-$5^i$.

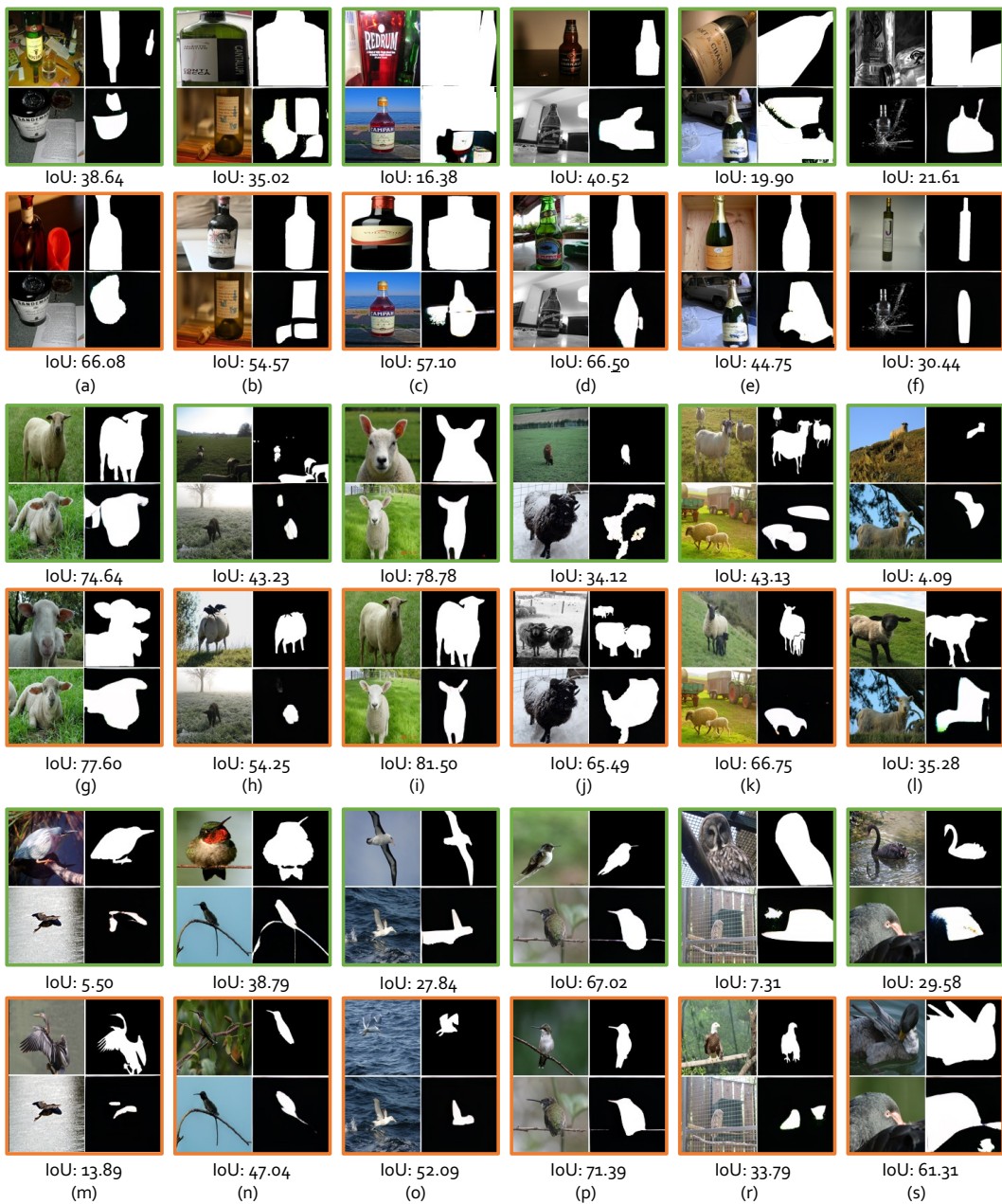

Figure 7: In-context examples, which are from the foreground segmentation task, retrieved by UnsupPR and SupPR. These grids show examples from the bottle, sheep, and bird categories.

## A.2 Single Object Detection

As shown in Fig. 12-13, we illustrate the in-context examples from the single object detection task. By comparing the in-context examples picked by UnsupPR and those picked by SupPR, we find the examples found by SupPR are more similar to the queries in terms of object pose (e.g., Fig. 12(f)), viewpoint (e.g., Fig. 12(r).

## A.3 Coloralization

As shown in Fig. 14-15, we illustrate the in-context examples from the colorization task. This task aims to map a gray-scale image to a color image. By comparing the in-context examples picked

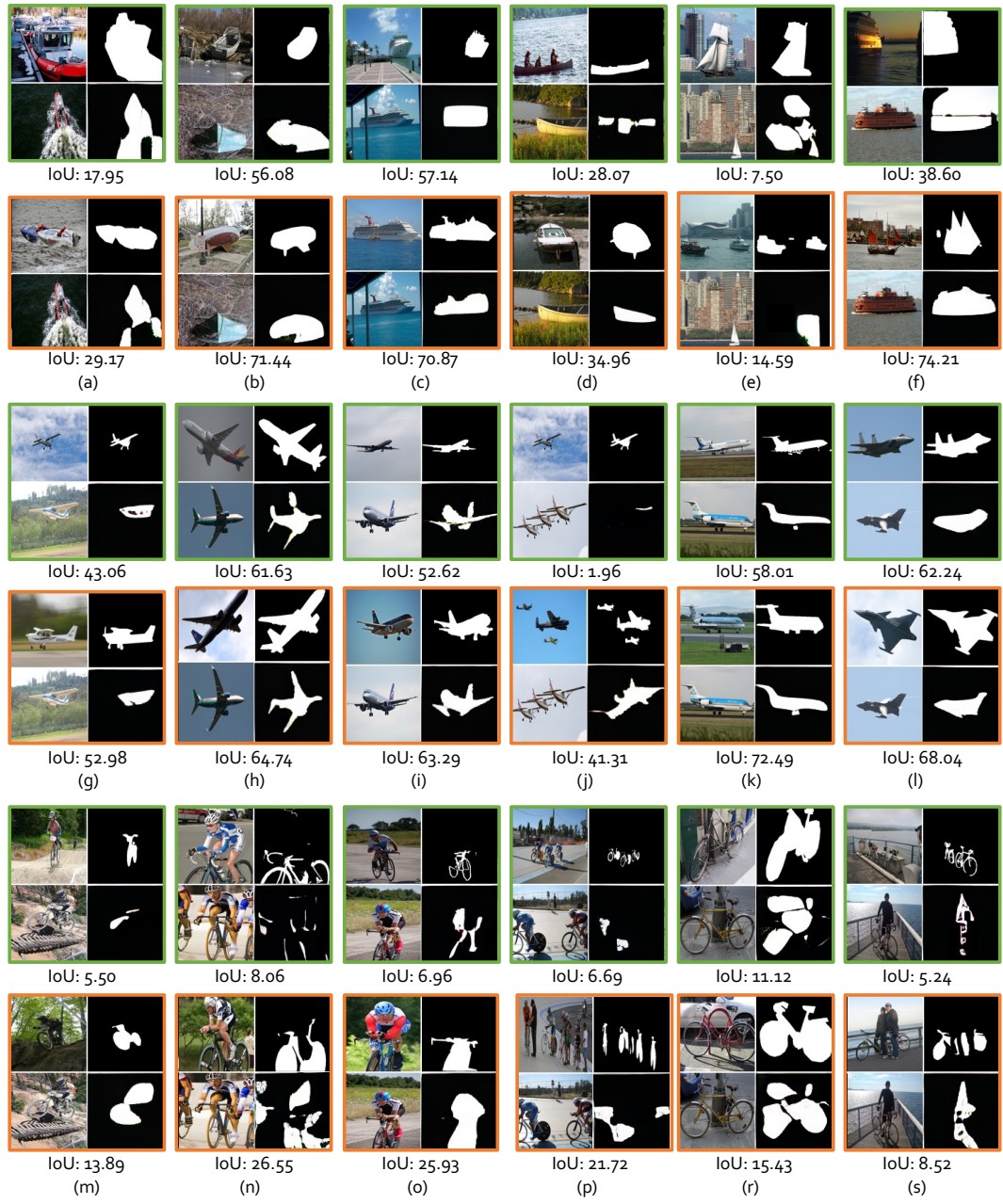

Figure 8: In-context examples, which are from the foreground segmentation task, retrieved by Un-supPR and SupPR. These grids show examples from the boat, airplane, and bicycle categories.

by UnsupPR and those picked by SupPR, we find the ground truth images of examples found by SupPR are more similar to that of the queries in terms of image style, *e.g.* the background color (e.g., Fig. 14(g)(h)).

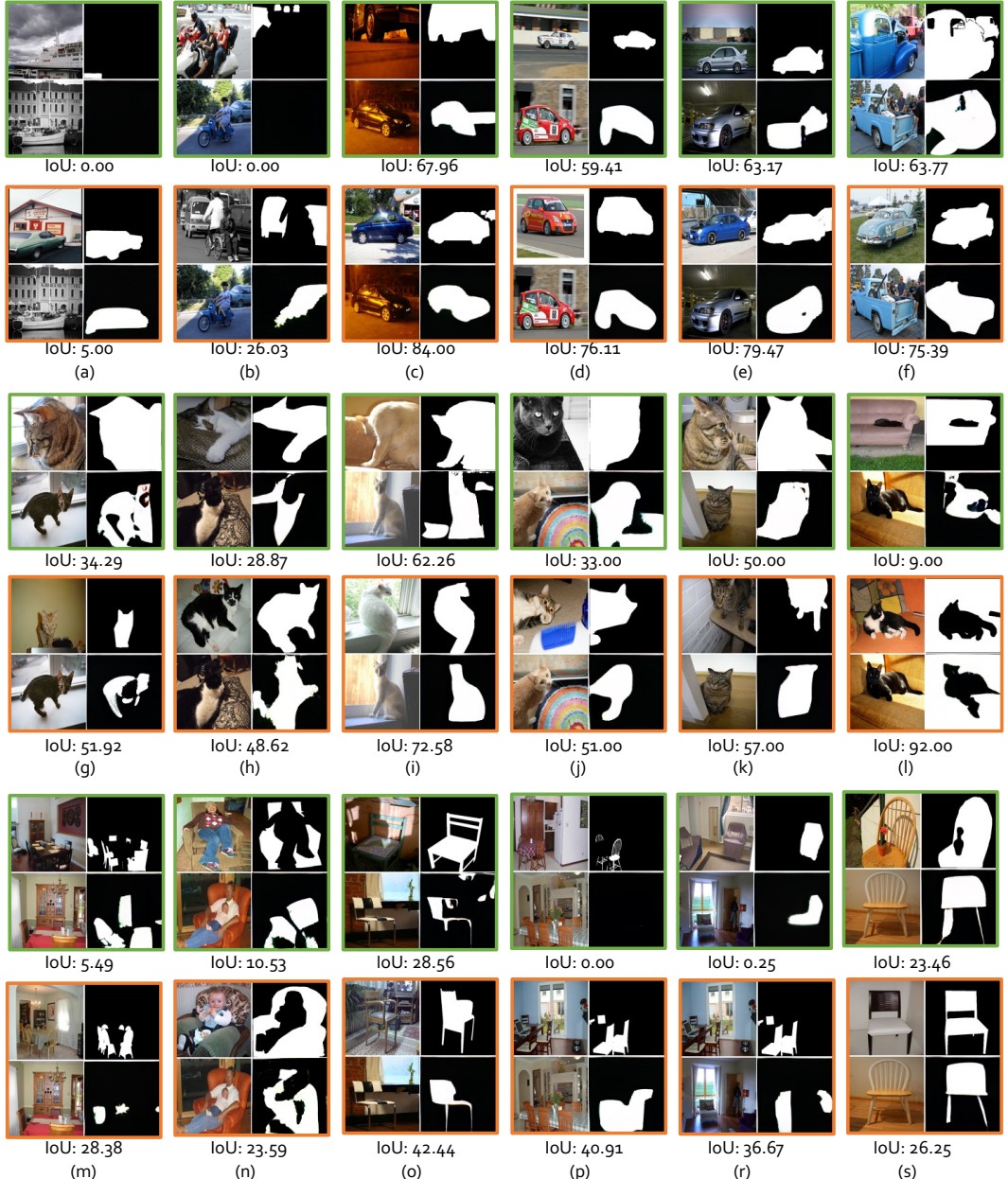

Figure 9: In-context examples, which are from the foreground segmentation task, retrieved by UnsupPR and SupPR. These grids show examples from the car, cat, and chair categories.

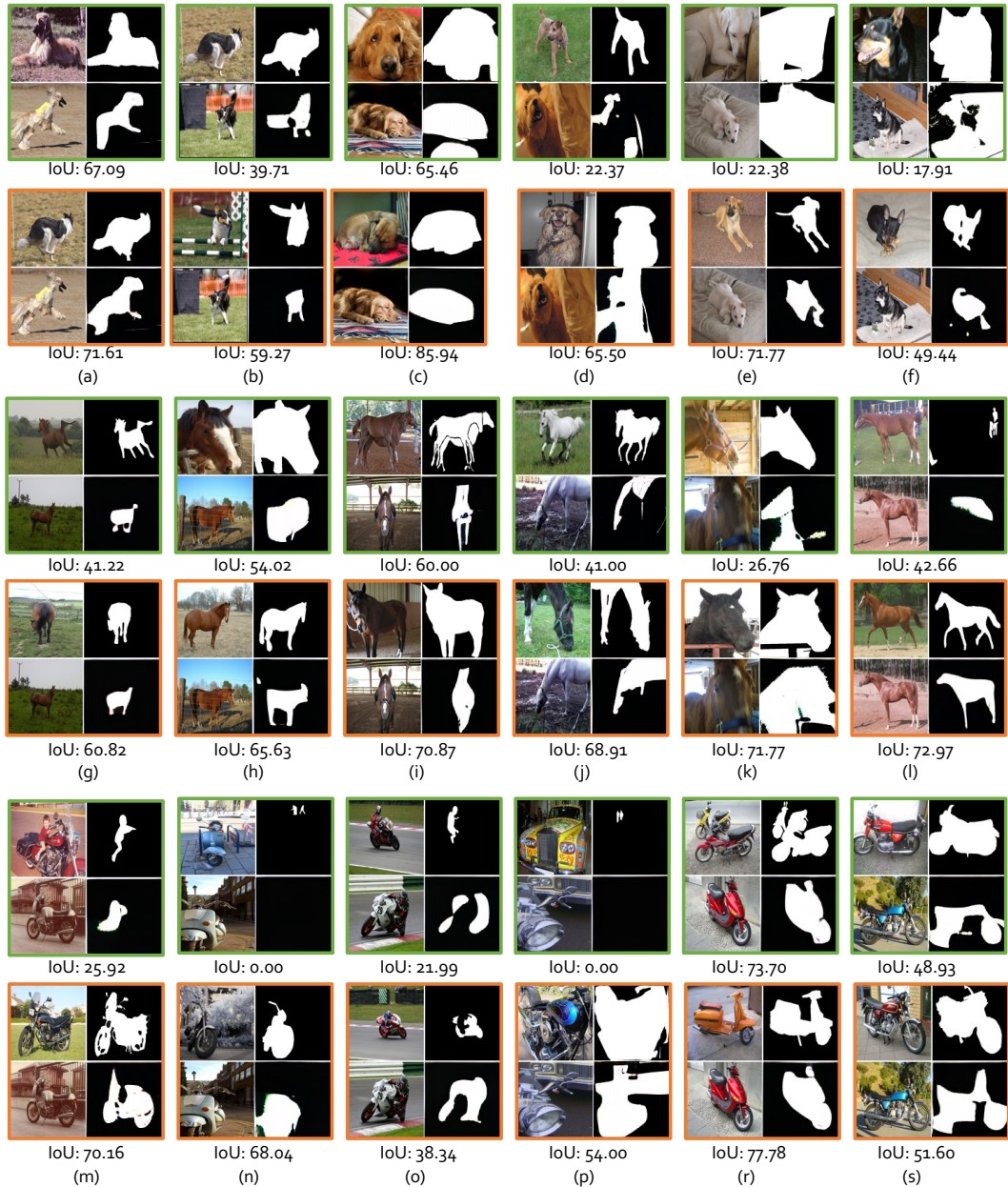

Figure 10: In-context examples, which are from the foreground segmentation task, retrieved by UnsupPR and SupPR. These grids show examples from the dog, horse, and motorbike categories.

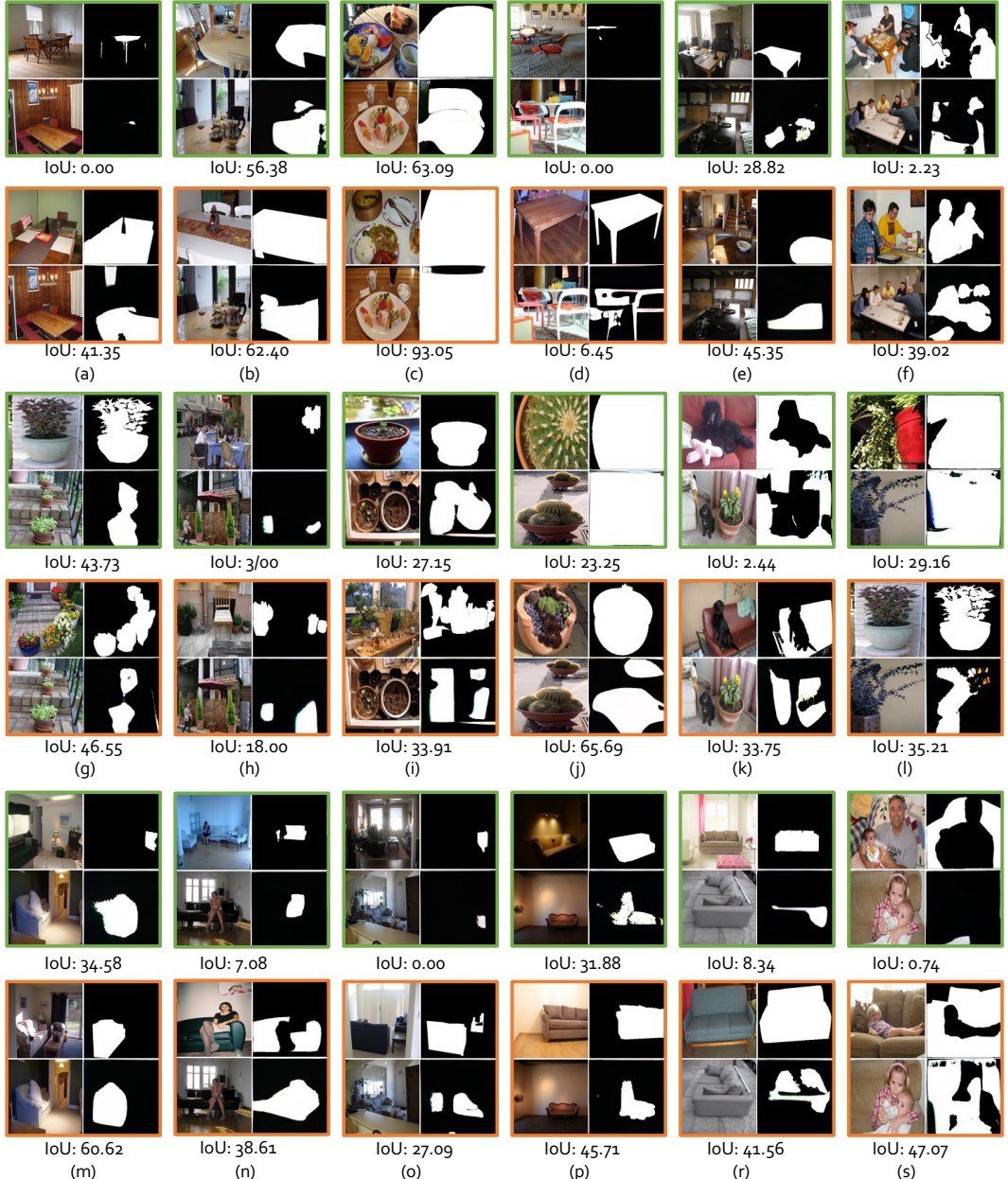

IoU: 0.00    IoU: 56.38    IoU: 63.09    IoU: 0.00    IoU: 28.82    IoU: 2.23

IoU: 41.35    IoU: 62.40    IoU: 93.05    IoU: 6.45    IoU: 45.35    IoU: 39.02

(a)    (b)    (c)    (d)    (e)    (f)

IoU: 43.73    IoU: 3/00    IoU: 27.15    IoU: 23.25    IoU: 2.44    IoU: 29.16

IoU: 46.55    IoU: 18.00    IoU: 33.91    IoU: 65.69    IoU: 33.75    IoU: 35.21

(g)    (h)    (i)    (j)    (k)    (l)

IoU: 34.58    IoU: 7.08    IoU: 0.00    IoU: 31.88    IoU: 8.34    IoU: 0.74

IoU: 60.62    IoU: 38.61    IoU: 27.09    IoU: 45.71    IoU: 41.56    IoU: 47.07

(m)    (n)    (o)    (p)    (r)    (s)

Figure 11: In-context examples, which are from the foreground segmentation task, retrieved by UnsupPR and SupPR. These grids show examples from the table, plant, and sofa categories.

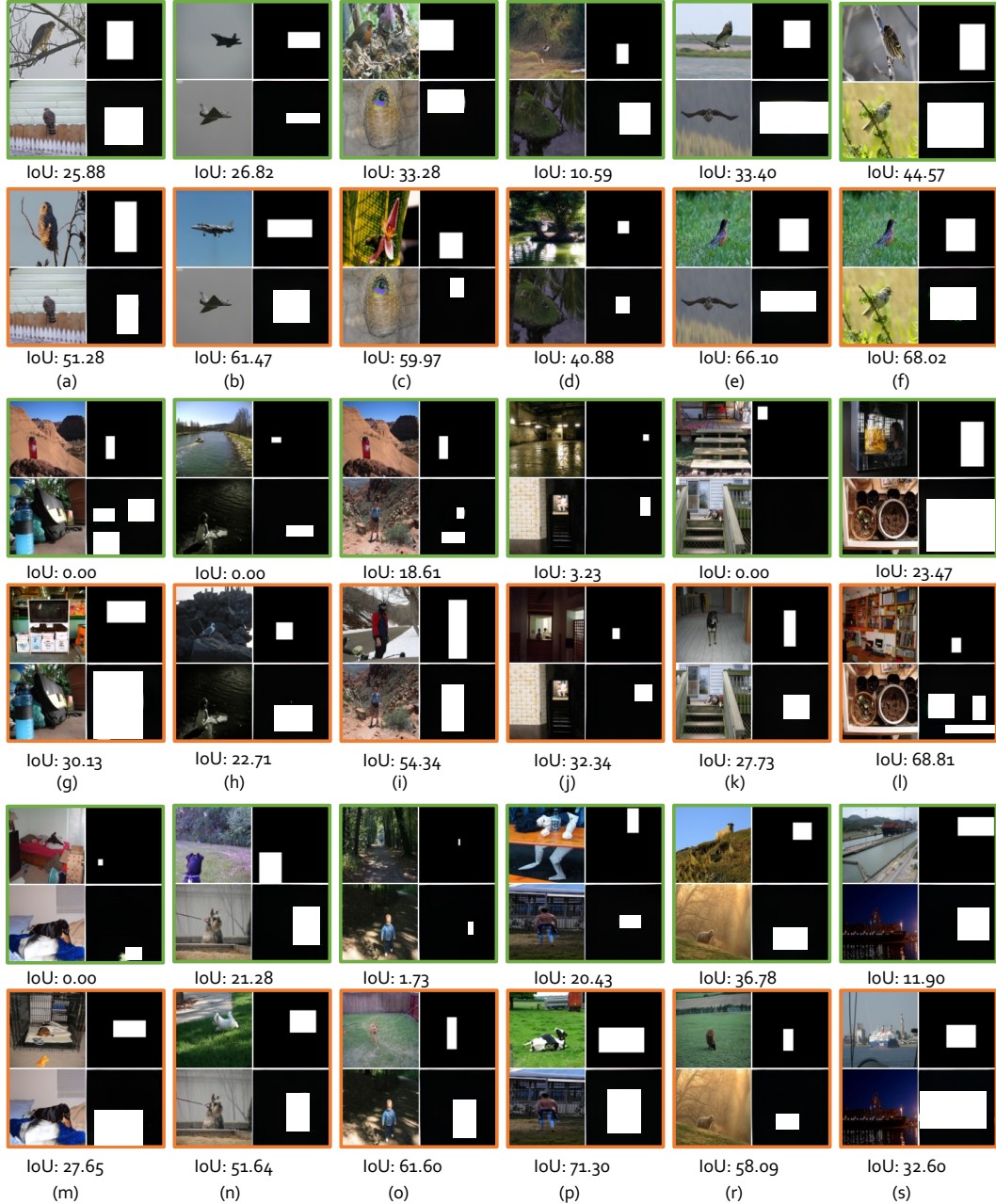

Figure 12: In-context examples, which are from the single object detection task, retrieved by Un-supPR and SupPR. We find the examples found by SupPR are more similar to the queries in terms of object pose (e.g., (f)), viewpoint (e.g., (r))

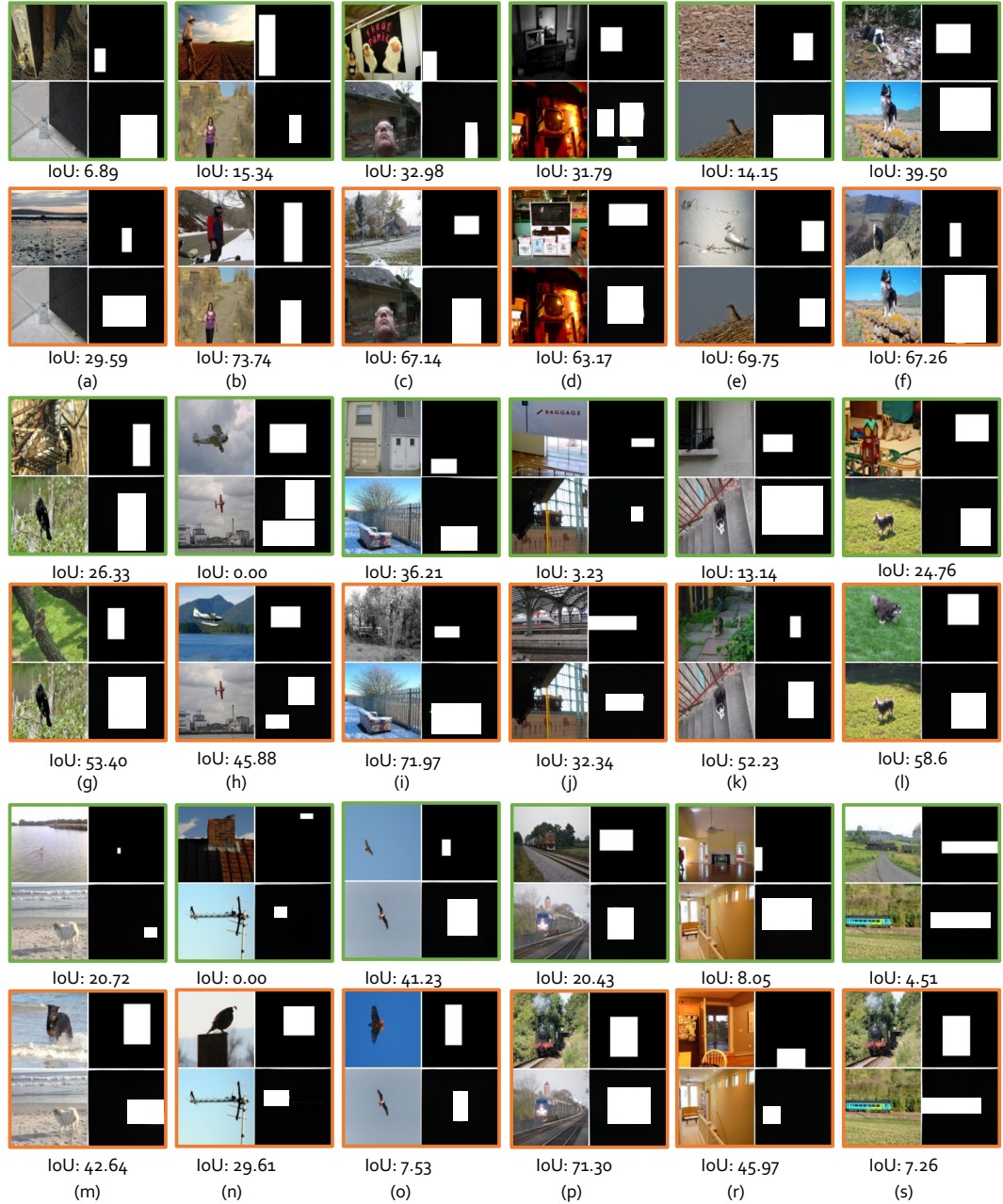

Figure 13: In-context examples, which are from the single object detection task, retrieved by Un-supPR and SupPR. We find the examples found by SupPR are more similar to the queries in terms of object pose (e.g., (l)), viewpoint (e.g., (m))

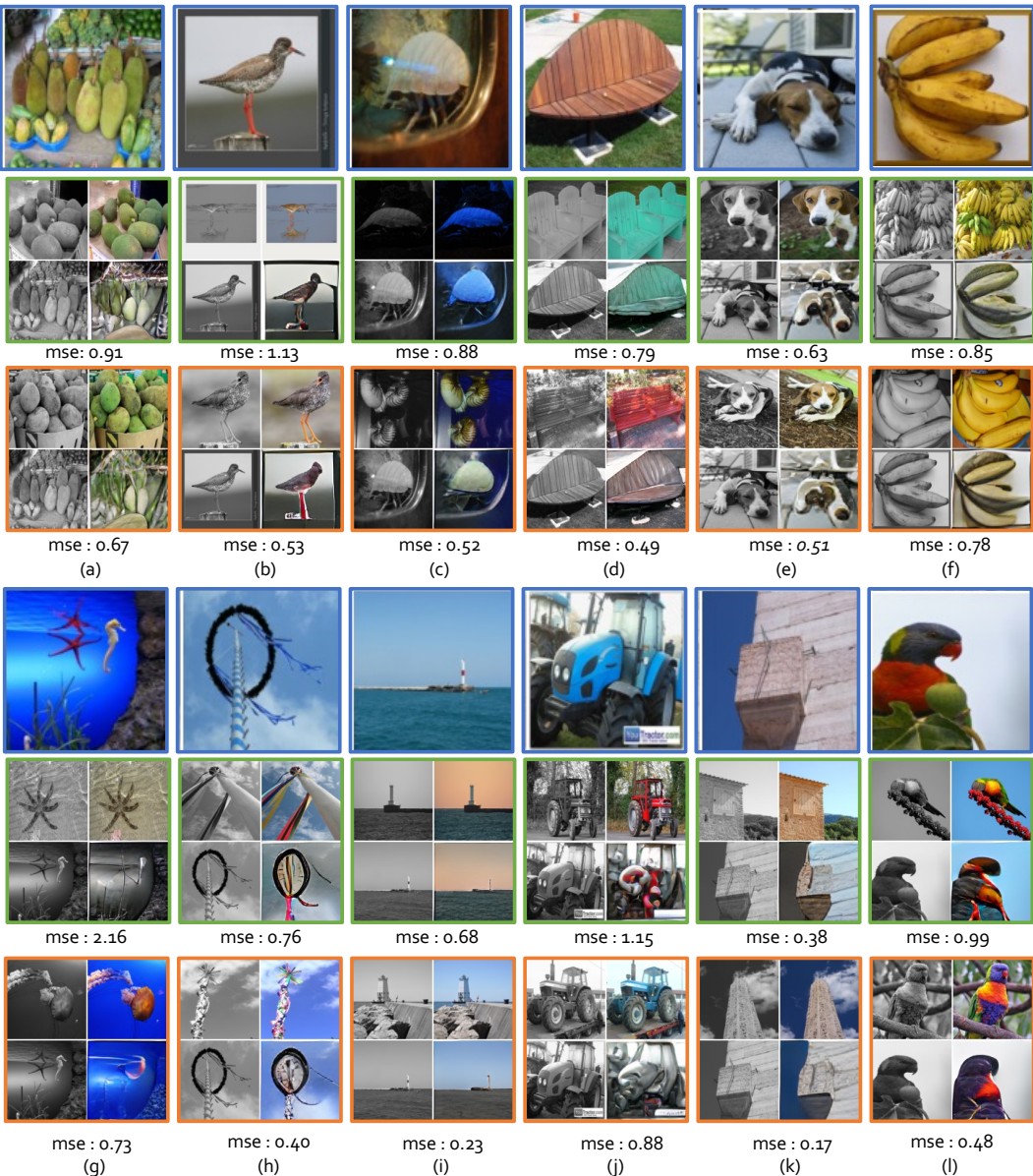

Figure 14: In-context examples, which are from the colorization task, retrieved by UnsupPR and SupPR. We also show the ground truth of the query image. The query image is the gray-scale version of its ground truth. The ground truth images of the in-context examples found by SupPR are more similar than those found by UnsupPR to the ground truth images of queries in terms of image style, *e.g.* the background color (g).

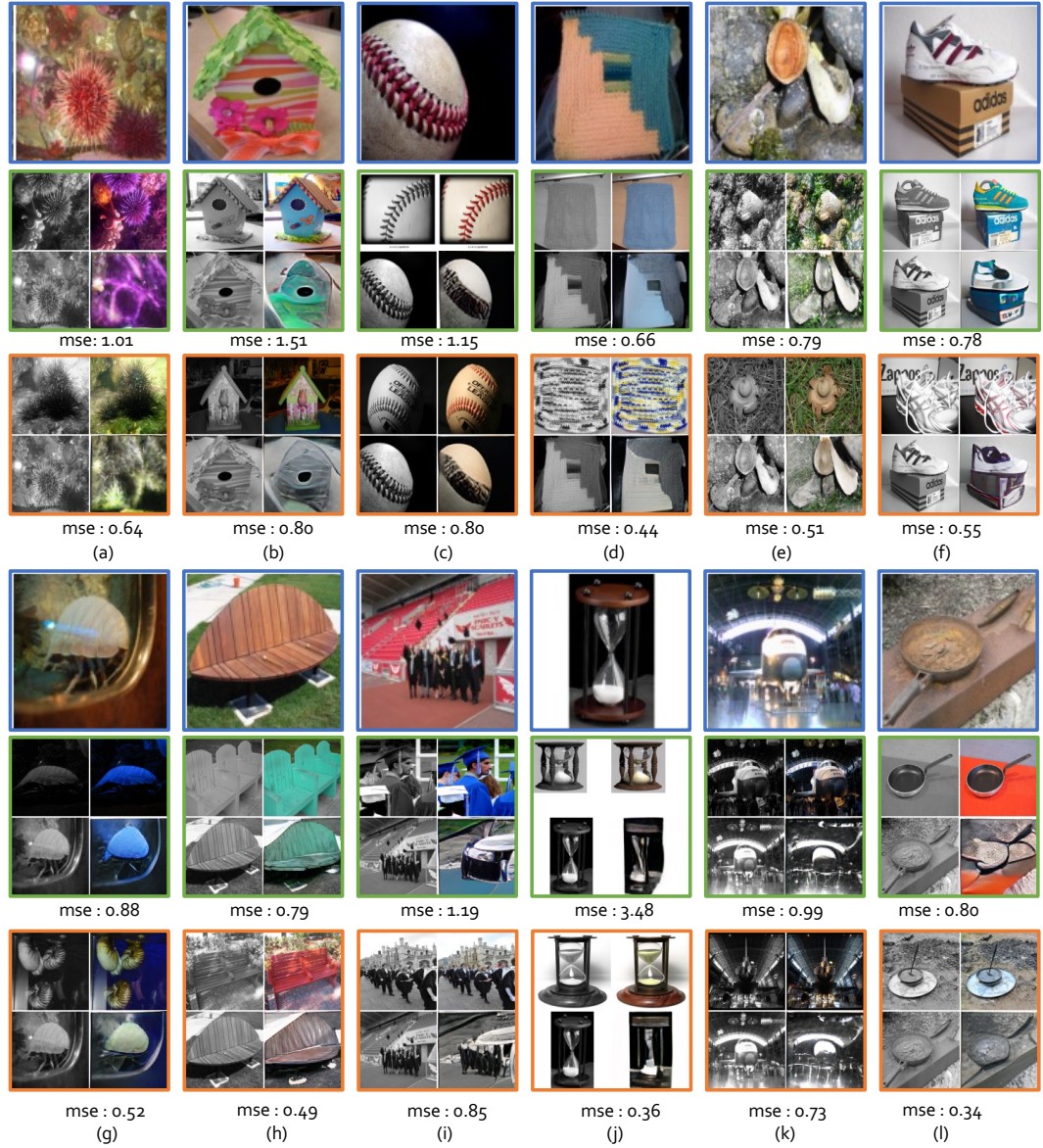

Figure 15: In-context examples, which are from the colorization task, retrieved by UnsupPR and SupPR. We also show the ground truth of the query image. The query image is the gray-scale version of its ground truth. The ground truth images of the in-context examples found by SupPR are more similar than those found by UnsupPR to the ground truth images of queries in terms of image style, *e.g.* the background color (h).

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
