# What Makes Good Examples for Visual In-Context Learning?

## A  Illustration of In-context Examples

In the supplementary material, we illustrate more in-context learning results of foreground segmentation, single object detection, and colorization tasks.

### A.1  Foreground Segmentation

The main paper presents the in-context examples from the person and cow categories. In the supplementary, as shown in Fig. 1-6, we present examples from the remained 18 categories in Pascal-5$^i$.

### A.2  Single Object Detection

As shown in Fig. 7-8, we illustrate the in-context examples from the single object detection task. By comparing the in-context examples picked by UnsupPR and those picked by SupPR, we find the examples found by SupPR are more similar to the queries in terms of object pose (e.g., Fig. 7(f)), viewpoint (e.g., Fig. 7(r).

### A.3  Coloralization

As shown in Fig. 9-10, we illustrate the in-context examples from the colorization task. This task aims to map a gray-scale image to a color image. By comparing the in-context examples picked by UnsupPR and those picked by SupPR, we find the ground truth images of examples found by SupPR are more similar to that of the queries in terms of image style, *e.g.* the background color (e.g., Fig. 9(g)(h)).

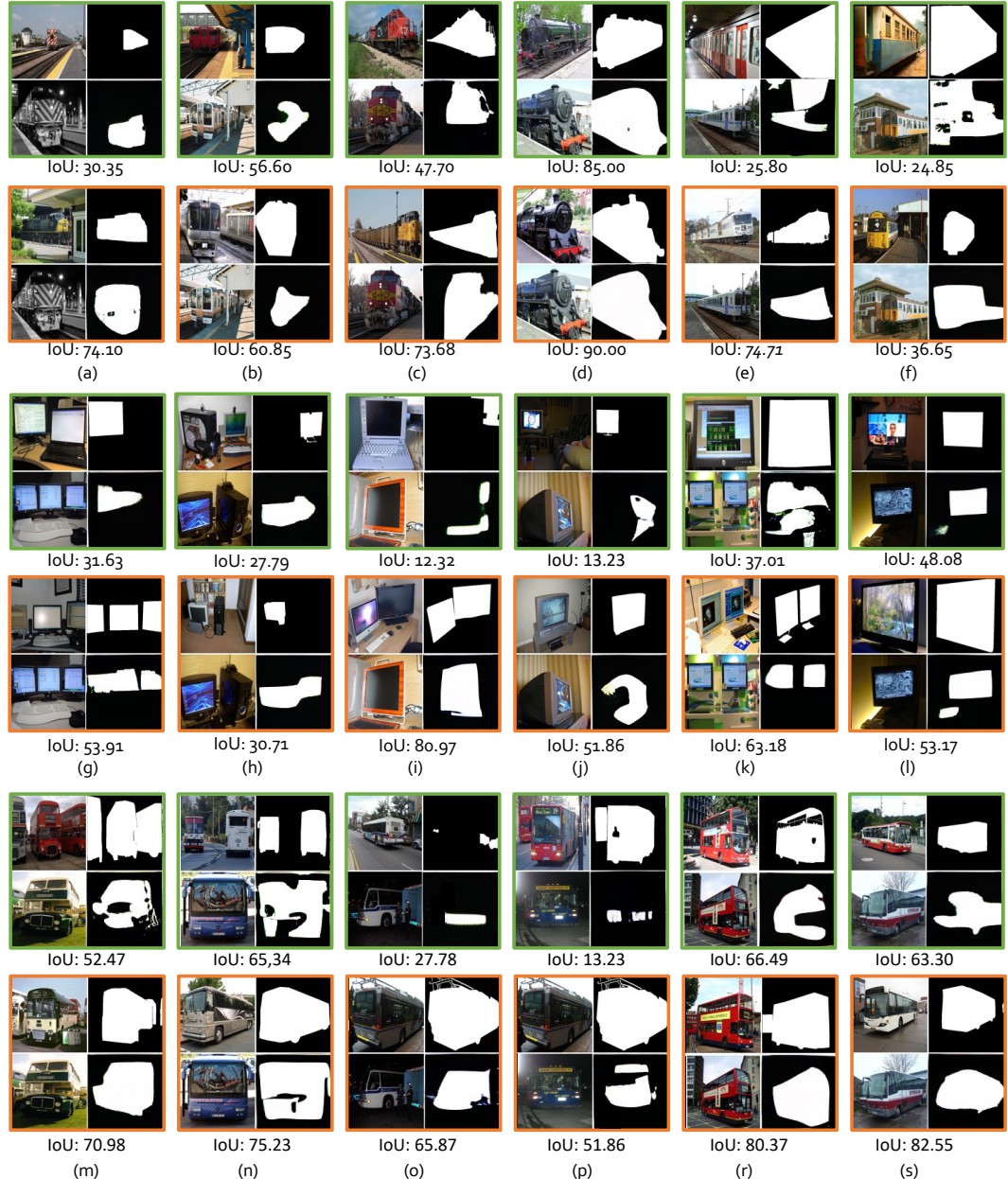

Figure 1: In-context examples, which are from the foreground segmentation task, retrieved by Un-supPR and SupPR. These grids show examples from the train, tv, and bus categories.

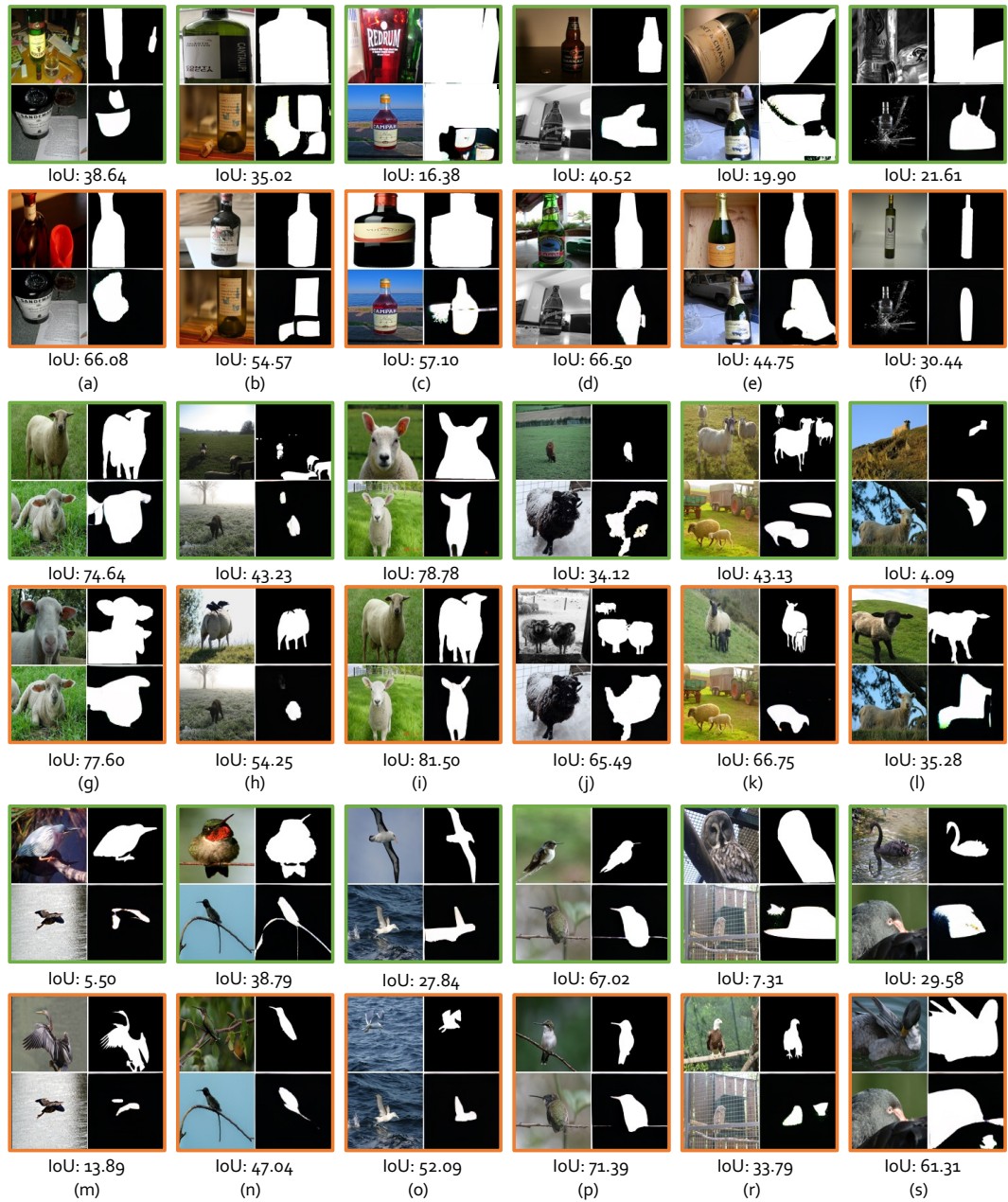

IoU: 38.64    IoU: 35.02    IoU: 16.38    IoU: 40.52    IoU: 19.90    IoU: 21.61

IoU: 66.08    IoU: 54.57    IoU: 57.10    IoU: 66.50    IoU: 44.75    IoU: 30.44

(a)      (b)      (c)      (d)      (e)      (f)

IoU: 74.64    IoU: 43.23    IoU: 78.78    IoU: 34.12    IoU: 43.13    IoU: 4.09

IoU: 77.60    IoU: 54.25    IoU: 81.50    IoU: 65.49    IoU: 66.75    IoU: 35.28

(g)      (h)      (i)      (j)      (k)      (l)

IoU: 5.50    IoU: 38.79    IoU: 27.84    IoU: 67.02    IoU: 7.31    IoU: 29.58

IoU: 13.89    IoU: 47.04    IoU: 52.09    IoU: 71.39    IoU: 33.79    IoU: 61.31

(m)      (n)      (o)      (p)      (r)      (s)

Figure 2: In-context examples, which are from the foreground segmentation task, retrieved by Un-supPR and SupPR. These grids show examples from the bottle, sheep, and bird categories.

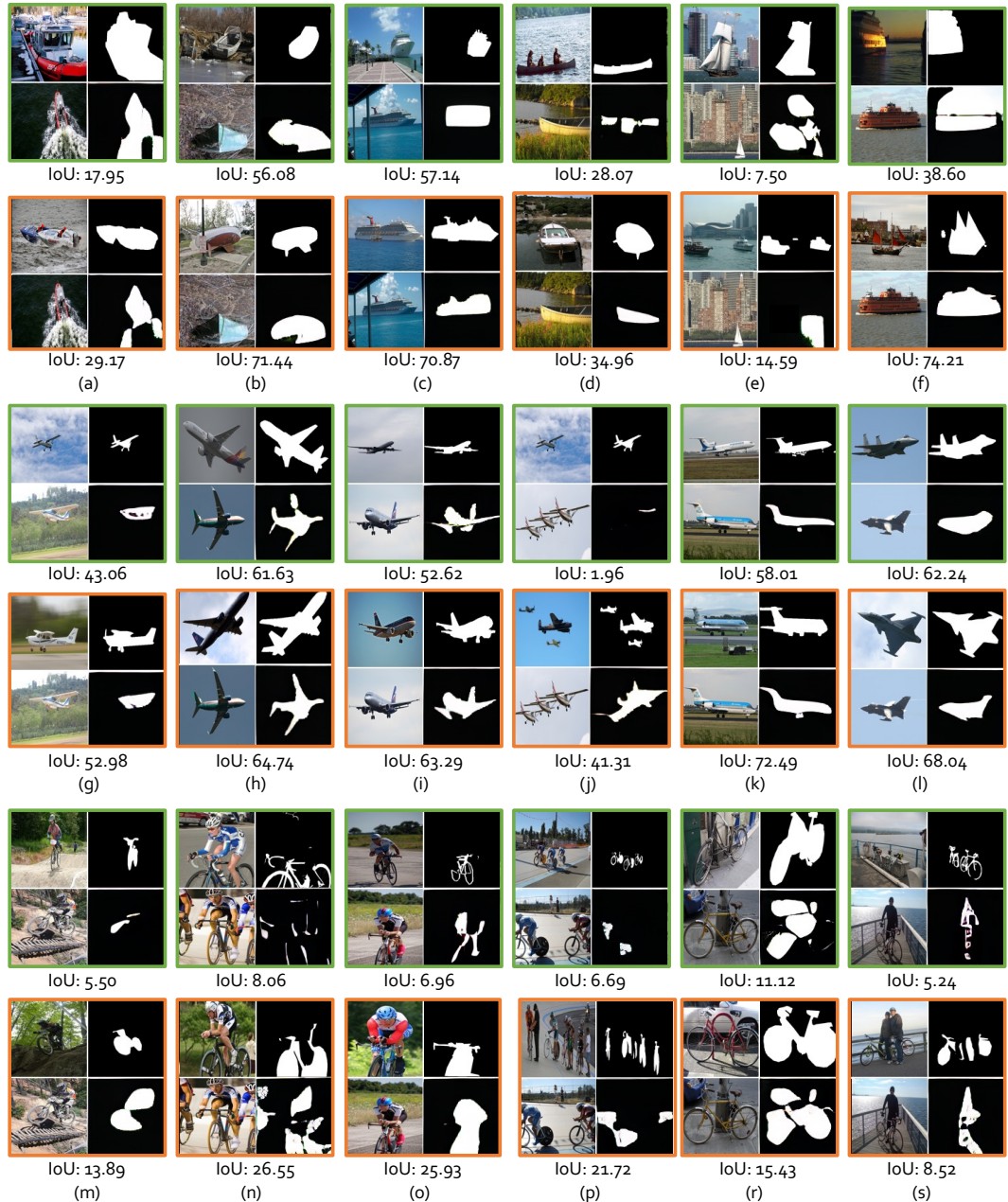

Figure 3: In-context examples, which are from the foreground segmentation task, retrieved by Un-supPR and SupPR. These grids show examples from the boat, airplane, and bicycle categories.

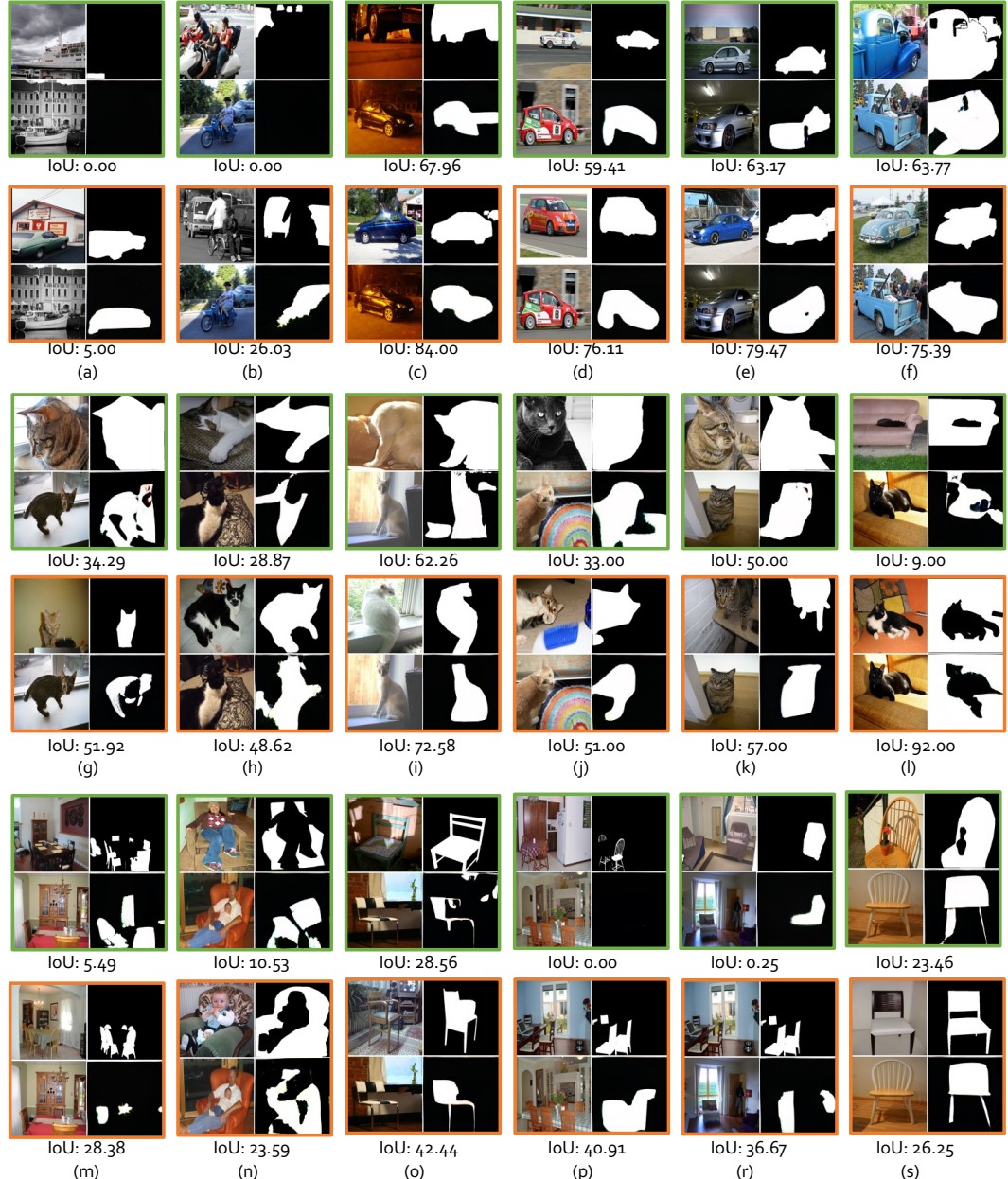

Figure 4: In-context examples, which are from the foreground segmentation task, retrieved by UnsupPR and SupPR. These grids show examples from the car, cat, and chair categories.

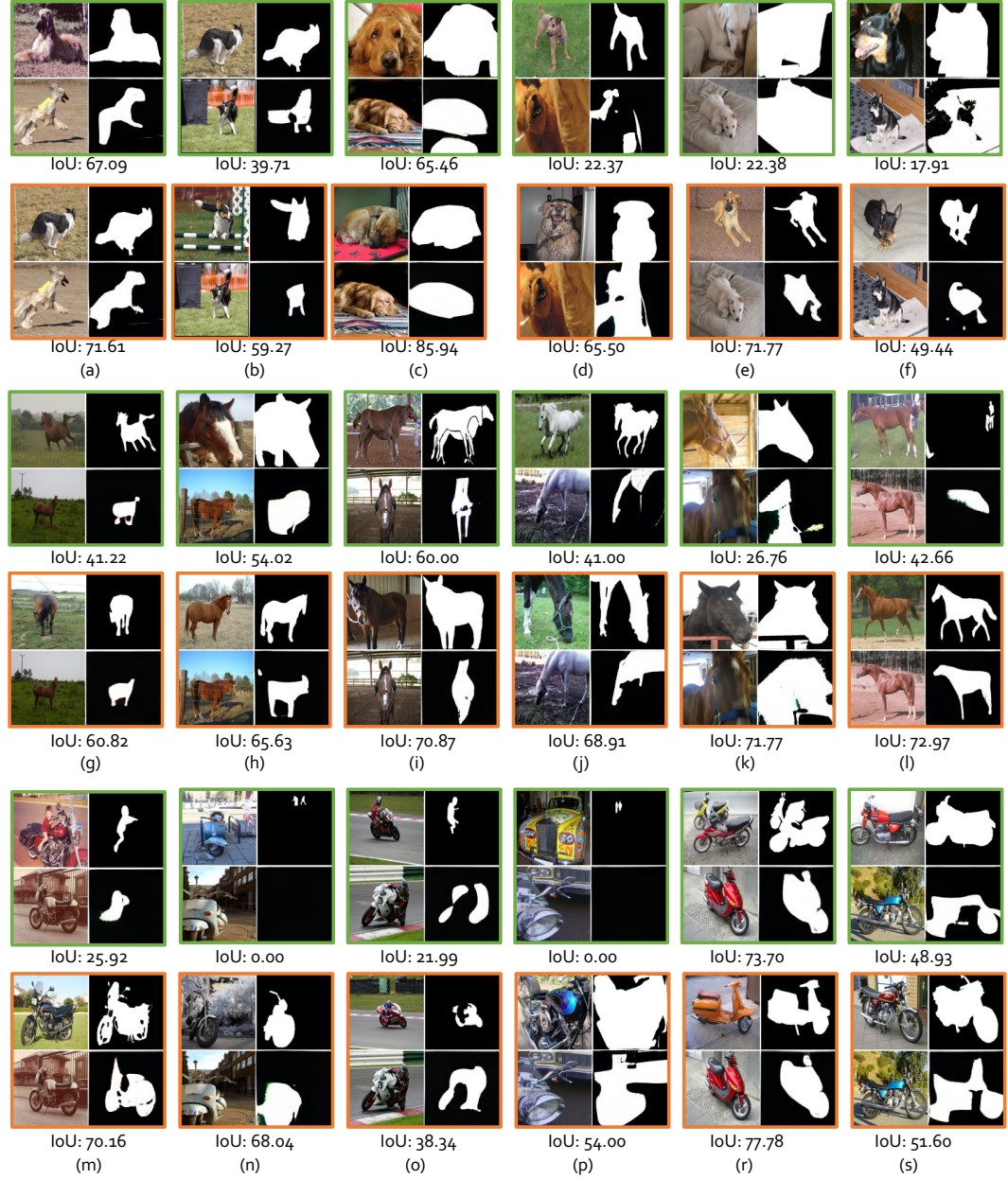

Figure 5: In-context examples, which are from the foreground segmentation task, retrieved by UnsupPR and SupPR. These grids show examples from the dog, horse, and motorbike categories.

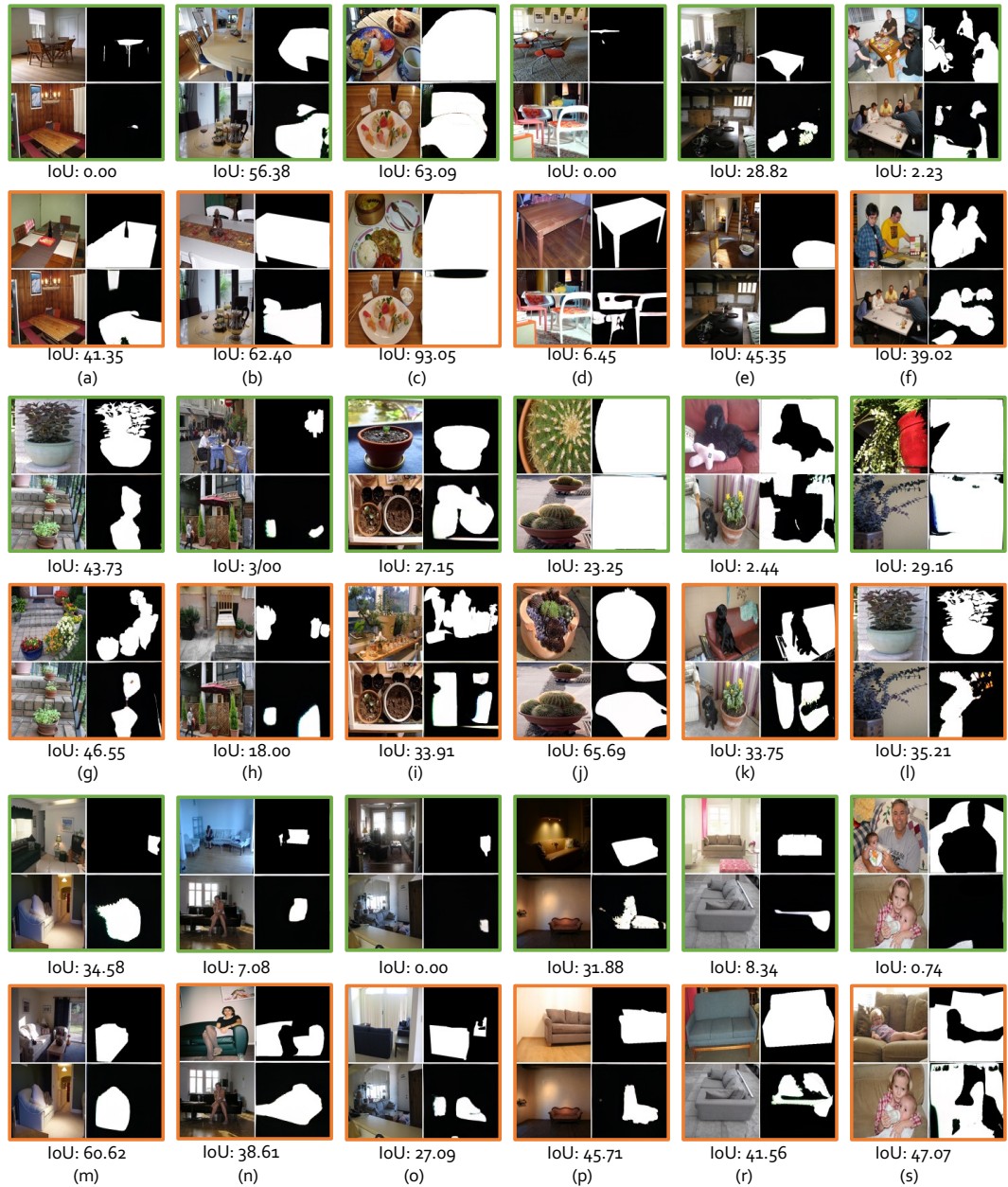

Figure 6: In-context examples, which are from the foreground segmentation task, retrieved by UnsupPR and SupPR. These grids show examples from the table, plant, and sofa categories.

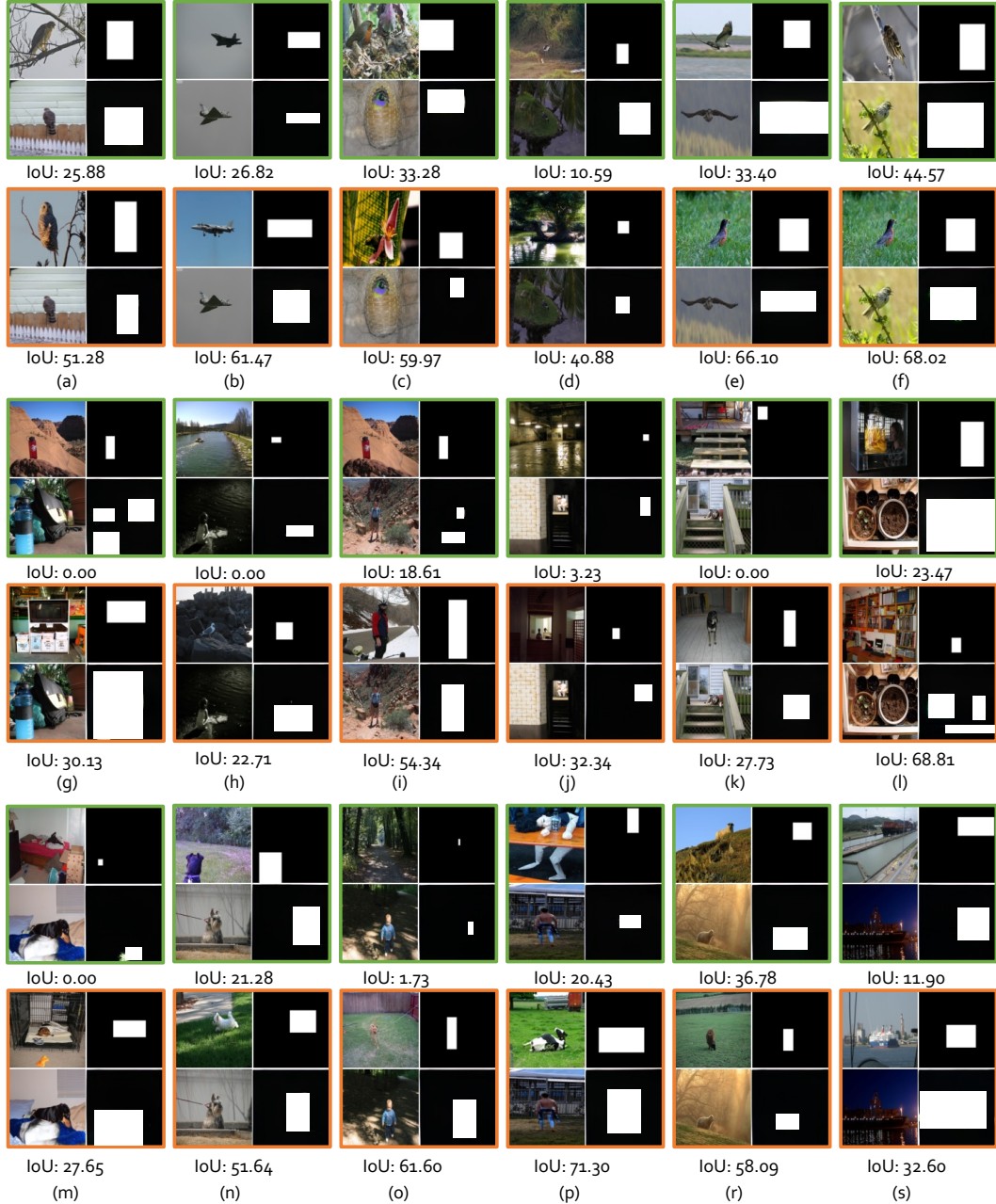

Figure 7: In-context examples, which are from the single object detection task, retrieved by Un-supPR and SupPR. We find the examples found by SupPR are more similar to the queries in terms of object pose (e.g., (f)), viewpoint (e.g., (r))

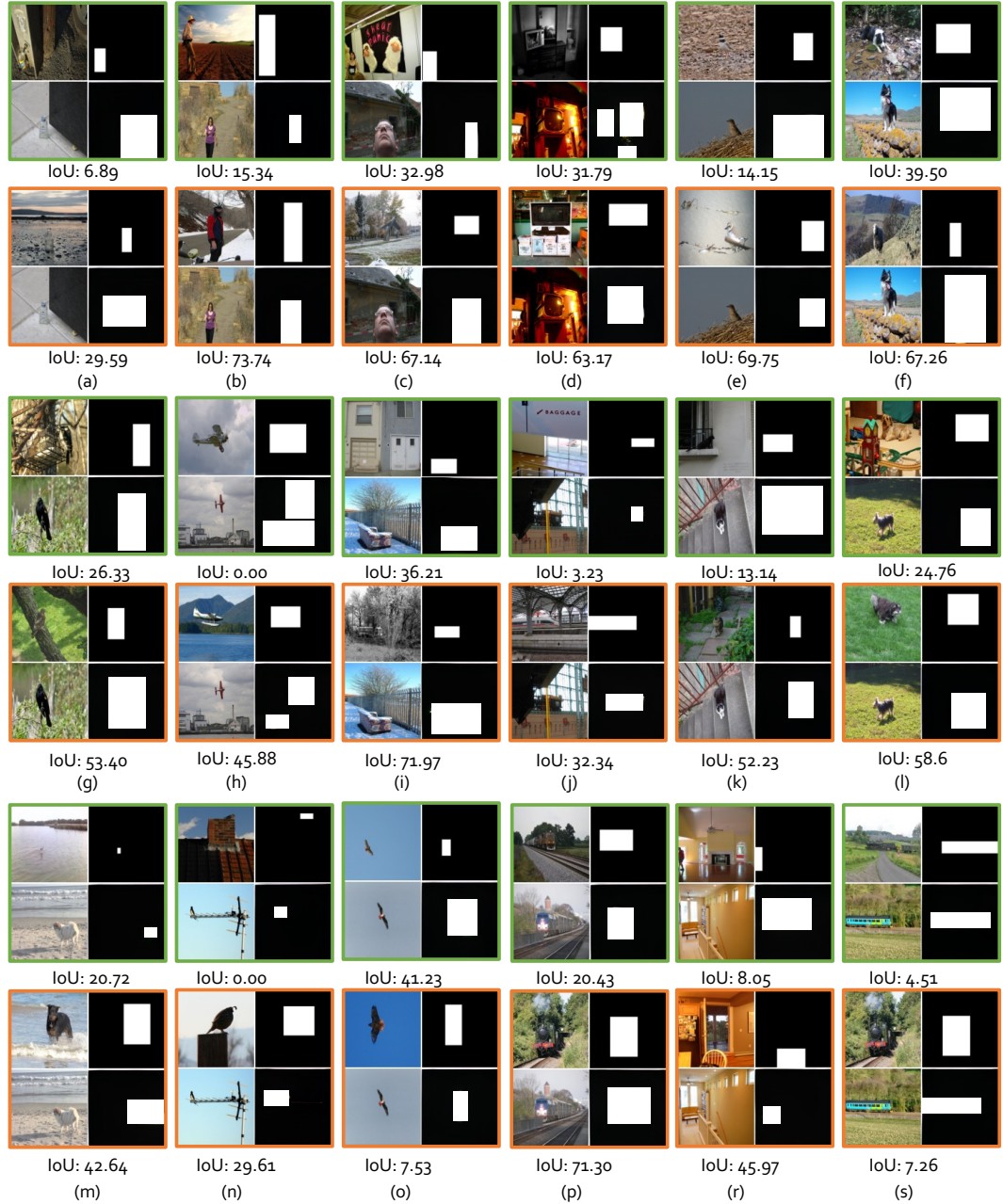

Figure 8: In-context examples, which are from the single object detection task, retrieved by Un-supPR and SupPR. We find the examples found by SupPR are more similar to the queries in terms of object pose (e.g., (l)), viewpoint (e.g., (m))

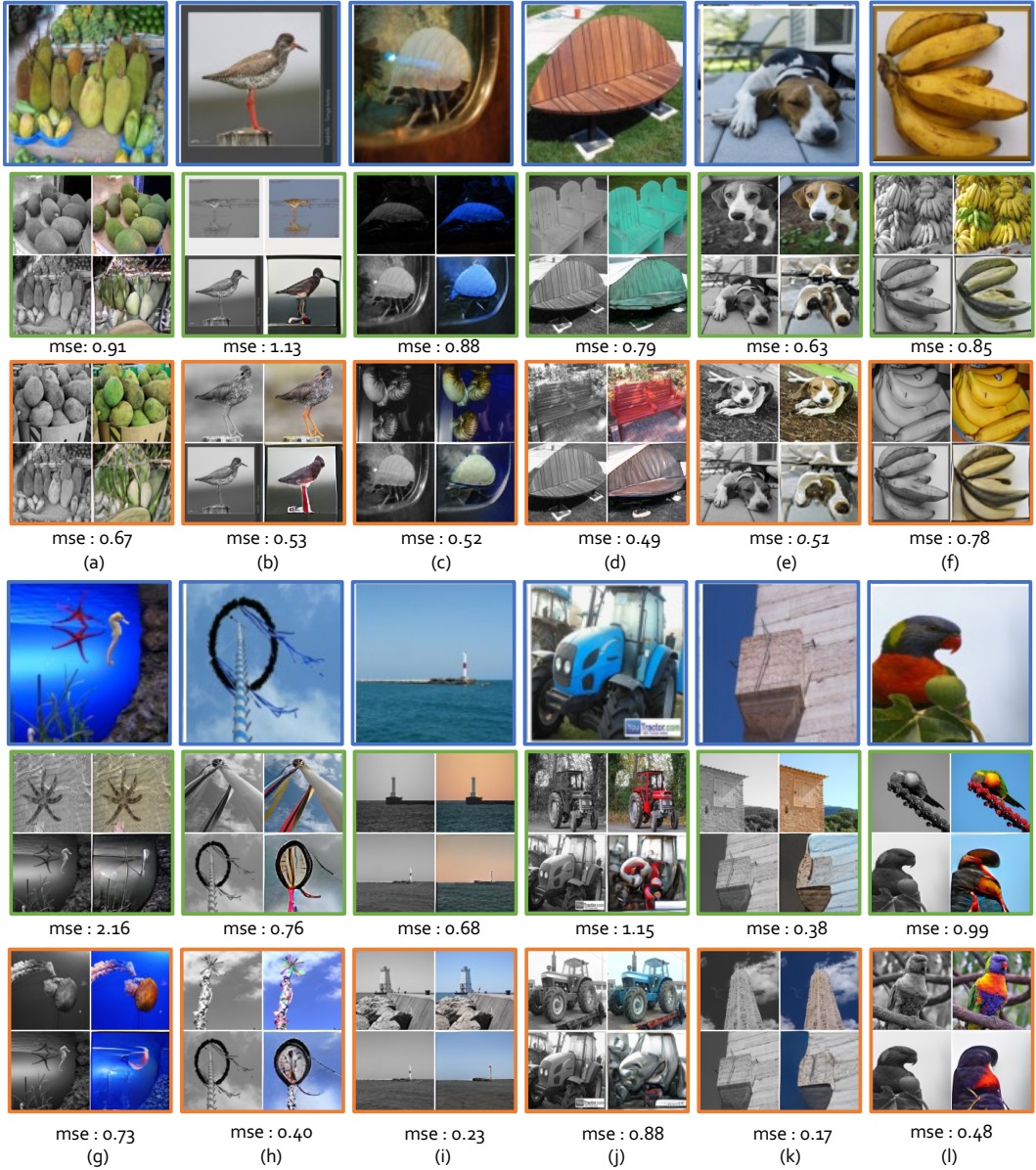

Figure 9: In-context examples, which are from the colorization task, retrieved by UnsupPR and SupPR. We also show the ground truth of the query image. The query image is the gray-scale version of its ground truth. The ground truth images of the in-context examples found by SupPR are more similar than those found by UnsupPR to the ground truth images of queries in terms of image style, *e.g.* the background color (g).

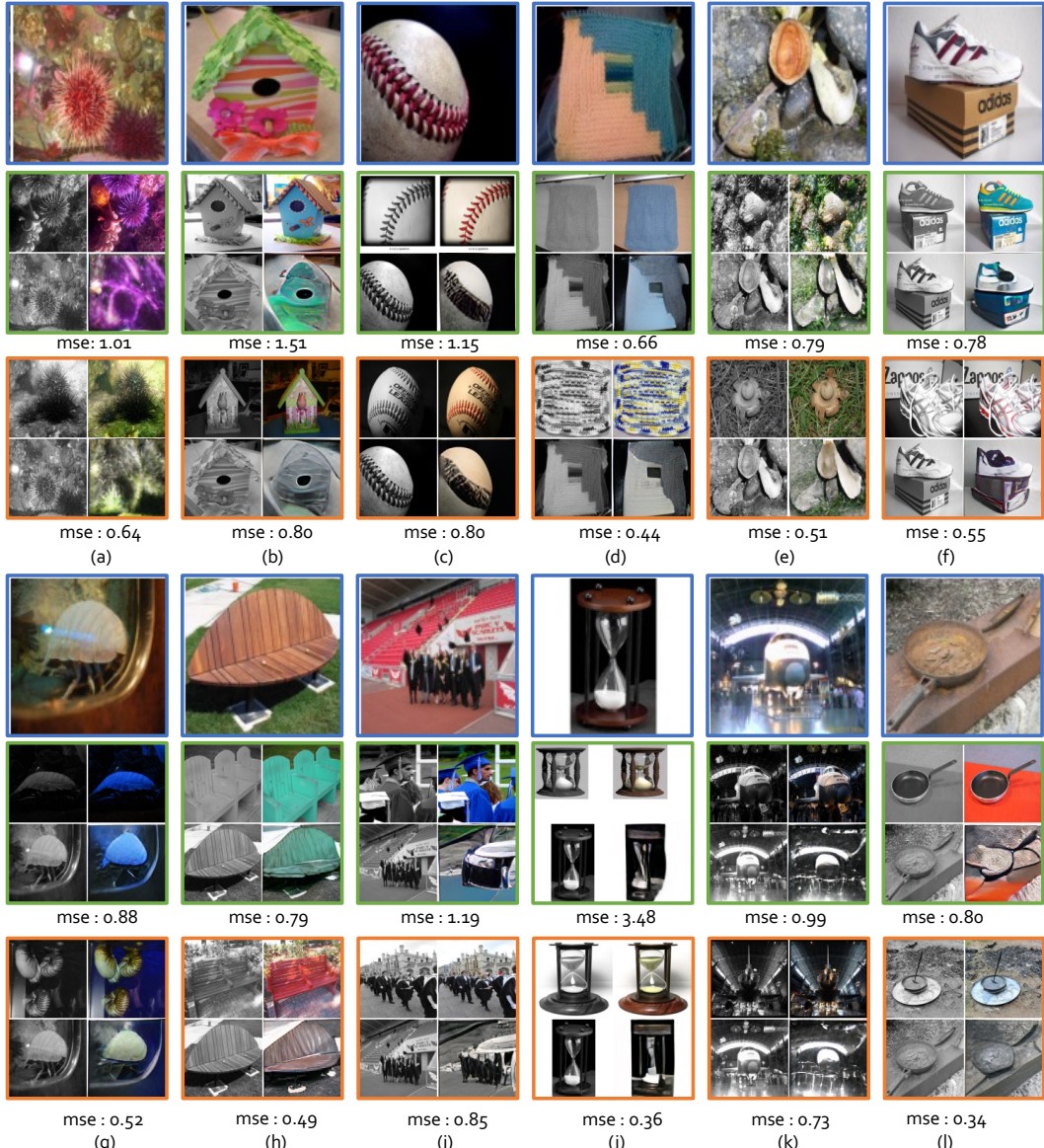

Figure 10: In-context examples, which are from the colorization task, retrieved by UnsupPR and SupPR. We also show the ground truth of the query image. The query image is the gray-scale version of its ground truth. The ground truth images of the in-context examples found by SupPR are more similar than those found by UnsupPR to the ground truth images of queries in terms of image style, *e.g.* the background color (h).