# OpenReview forum: "What Makes Good Examples for Visual In-Context Learning?"
_NeurIPS.cc/2023/Conference — NeurIPS 2023 poster_

### Official Review · Reviewer_vwKB · 2023-07-03

**Soundness:** 3 good
**Presentation:** 3 good
**Contribution:** 3 good
**Rating:** 7
**Confidence:** 4

**Summary:**

This paper studied in-context learning abilities of large vision models and finds downstream task performance to be highly sensitive to the choice of examples. They observe that the closer the in-context example is to the query, the better the performance. Since manually designing prompts would be time-intensive, they propose a (contrastive) supervised and an unsupervised version of a framework for prompt retrieval guided by a score function. They evaluate the proposed framework on the foreground segmentation, single object detection and image colorisation tasks.

**Strengths:**

- In-context learning in vision is a new and active area of research and methods for selecting better examples for in-context learning will likely be of interest to the computer vision community.
- The proposed method for prompt retrieval outperforms randomly selected examples on all tasks considered.
- Distribution shift results are interesting, suggesting that the supervised prompt retrieval method acquires in-context knowledge that is robust to distribution shift.
- The analysis and ablations are interesting and insightful.

**Weaknesses:**

- The finding that examples in the prompt should be semantically close to the test example is known from language and not that surprising.
- The proposed method is technically relatively straightforward compared to an average NeurIPS paper.
- The analysis is limited to one model for in-context learning and may not transfer to other models.


**Questions:**

1. It is surprising to me that the smallest gains are observed on the colorisation task and supervision is not helpful, despite the fact that both training and evaluation are done on ImageNet. Do you have any intuition for why this may be the case?
2. In Sec 3.3, you mention that 20% of the data is enough for good performance. Is this 20% of 50,000, i.e. 1000 images?
3. What is the intuition behind MAE in Table 5 performing worse than UnsupPR?

**Limitations:**

Yes.

---

> ### Author Rebuttal · Authors · 2023-08-08
>
> We thank the reviewer for the feedback and comments. We respond to the concerns below:
>
> **Q1**: Semantically close to the test example is known from language and not that surprising.
>
> **A1**: Yes, we acknowledge the parallels between some of our observations and those from the natural language domain. However, most of our study presents findings that are significantly novel to the computer vision community. For example, while the concept of semantic similarity might seem intuitive initially, the specific attributes that constitute this similarity in the context of computer vision remain largely unexplored. As highlighted in Figure 3, elements such as background, pose, appearance, and viewpoint play crucial roles.
>
> Furthermore, our study identified differences between language and vision models. Notably, while in-context learning in language models shows sensitivity to the sequencing of in-context examples[1], such order doesn't significantly influence vision models. This underscores the unique challenges and considerations intrinsic to computer vision tasks.
>
> **Q2**: Technically relatively straightforward compared to an average NeurIPS paper.
>
> **A2**: Our proposed methodology, specifically for the SupPR, introduces an innovative approach to prompt retrieval that efficiently boosts the downstream capabilities of the pre-trained model without fine-tuning it. In addition, as shown in table.5, the superiority of prompt retrieval becomes apparent only when we meticulously develop an appropriate learning pipeline, such as our SupPR, which effectively leverages this feedback to enhance the downstream performance of the pre-trained model. We believe our prompt retrieval design is a beneficial contribution to the community.
>
> **Q3**: One model for in-context learning and may not transfer to other models
>
> **A3**: Thanks for your suggestion. Please refer to the **General Response** for the detailed explanation.
>
> **Q4**: Smallest gains are observed on the colorization task and supervision is not helpful
>
> **A4**: A possible explanation for this observation is that the current pre-trained inpainting model does not efficiently perform in-context learning on the colorization task (for a detailed illustration, please refer to Figures 14-15 in the supplementary material). If the pre-trained model's in-context learning ability for the colorization task is weak, we face difficulties in ranking positive and negative examples relative to a query image. This, in turn, impedes the effective training of a robust feature extractor for SupPR.
>
> **Q5**: 20% of the data is enough for good performance
>
> **A5**: Given that each data split comprises approximately 3,000 samples, this implies that a subset of 600 samples (20% of the data) is deemed sufficient for effective model performance.
>
> **Q6**: Intuition behind MAE in Table 5 performing worse than UnsupPR
>
> **A6**: The differential in performance is fundamentally attributed to the complexity of designing a robust feature extractor for SupPR. The strength of SupPR comes from our intellectual design rather than the supervision signal. Without such careful design, the SupPR can even perform worse than the UnsupPR.
>
> Moreover, this finding highlight that semantic similarity is a key factor in selecting an effective prompt. The inferiority performance of the fine-tuned MAE suggests that features optimized by foreground segmentation may be deficient in encoding semantic information. On the other hand, given CLIP's well-known ability in encoding semantic information, our UnsupPR is equally good at retrieving semantically similar examples.
>
> [1] Agrawal, Sweta, et al. "In-context examples selection for machine translation." arXiv preprint arXiv:2212.02437 (2022).

---

> > ### Author Response · Authors · 2023-08-14
> > **Follow-up**
> >
> > Dear Reviewer,
> >
> > In our rebuttal:
> >
> > 1) We highlight the insight found in our paper. We claim that most of our study presents findings that are significantly novel to the computer vision community. And we indicate that the Table.5 in our paper shows that, despite our method is straightforword, the superiority of our method becomes apparent only when we meticulously develop an appropriate learning pipeline.
> >
> > 2) We conduct more experiments showing that method has demonstrated easy generalization to visual-language tasks such as image captioning and VQA.
> >
> > 3) We clearly answer your question about the detail of our paper.
> >
> > We would love your feedback on whether our answer has solved your concern or if you have further questions.

---

> > > ### Comment · Reviewer_vwKB · 2023-08-14
> > > **Response**
> > >
> > > Thank you for submitting the rebuttal. After reading your response and the general response where you show your method is general and applicable to VQA and captioning tasks, I've decided to increase my score to 7.

---

> > > > ### Author Response · Authors · 2023-08-14
> > > > **Thanks for raising your score!**
> > > >
> > > > Thank you for raising your score! We're glad our experiments on the VQA and captioning tasks addressed your concerns. We value your support for the paper's acceptance.

---

### Official Review · Reviewer_EQqT · 2023-07-04

**Soundness:** 3 good
**Presentation:** 4 excellent
**Contribution:** 3 good
**Rating:** 6
**Confidence:** 4

**Summary:**

The paper identifies that visual prompting is sensitive to the choice of input-output example(s). To address this, the authors propose a retrieval framework to better select the examples. The authors propose supervised and unsupervised retrieval approaches which significantly improve the performance compared to random selection.


**Strengths:**

- Through extensive empirical study, the authors demonstrate the role of input-output examples for visual prompting.
- The authors present two different retrieval strategies to choose the best visual prompting example and find that both approaches are superior to random choice.
- The authors show that choosing the right example can also improve performance under distribution shifts.
- It is also nice that the retrieval similarity function is class-agnostic. But regardless, empirically it extends to three different tasks.

**Weaknesses:**

Overall, the main two weaknesses in my opinion are:
1. The assumption that we have a set of tagged examples to retrieve from.
2. The supervised/unsupervised similarity function is straightforward but currently very specific to mIOU similarity.

Minor:

3. I think there is a broader spectrum that can be analyzed for visual prompts.

Please see the questions for extended comments/feedback.

**Questions:**

1. The main weakness is the assumption that we have a set of tagged examples. In this case, why should we retrieve rather than use our tagged examples? e.g., train a new model overall tagged examples or use an ensemble? Perhaps the best way to address this would be to add another ensemble/supervised-trained baseline in Figure 5 (left). These baselines should utilize the same pool of examples.

2. The supervised/unsupervised similarity function is straightforward but currently very specific to mIOU similarity. Can we improve this by utilizing similarity in feature space? (e.g., using existing CLIP or DINO, etc?). To be fair, from the empirical results it seems like by optimizing mIOU there is somewhat improvement for colorization and single object detection as well.

Minor:

3. I think there is a broader spectrum that can be analyzed for visual prompts. E.g., using synthetic task data as examples, using examples from different classes, then examples from within classes as analyzed in the paper.

**Limitations:**

The authors clearly stated and discussed the limitations of this work.

---

> ### Author Rebuttal · Authors · 2023-08-08
>
> We thank the reviewer for the feedback and comments. We respond to the concerns below:
>
>
> **Q1**: The assumption that we have a set of tagged examples to retrieve from. Why should we retrieve rather than training our tagged examples?
>
> **A1**: It is essential to note that our method is fundamentally designed for scenarios where there are a certain amount of tagged examples, which aligns with the standard presumption of in-context learning.
>
> In addition, we conduct studies to demonstrate the advantages of prompt retrieval. The following table, referring to Figure 5 (left), compares full-set and 1% of the full-set scenarios. The first two columns compare the performance of our SupPR model with the fine-tuned MAE model (the supervised-trained baseline that might concern you). Importantly, SupPR outperforms the fine-tuned model, especially when training data is scarce, underlining the superiority of the prompt retrieval approach.
>
> Furthermore, to accentuate the potential of visual in-context learning via prompt retrieval, we present an upper-bound performance, achieved by manually selecting the best in-context samples from the training dataset for each query. The noticeable performance difference between this upper bound and the fine-tuned MAE model highlights the potential and efficacy of prompt retrieval. If optimized, it can substantially enhance the performance of visual in-context learning, spurring further research in this field.
> |            | Fine-tune (MAE) | SupPR | Upper bound |
> |------------|-----------------|-------|-------------|
> | Seg.(mIoU) 1% full-set| 20.15           | 30.22 | 34.53       |
> | Seg.(mIoU) full-set| 34.98           | 35.56 | 40.23       |
>
>
>
> **Q2**: Supervised/unsupervised similarity function is straightforward but currently very specific to mIOU
>
> **A2**: Our similarity function indeed uses cosine similarity between two CLIP embeddings in the feature space. And, it's might be a misconception that it's inherently specific to mIOU. We emphasize that our methods have applicability beyond mIOU, including diverse generative tasks like image caption generation and VQA. Please refer to the **General Response** section, demonstrating the extendability of our method to other vision in-context learning tasks (image captioning and VQA), not limited to inpainting.
>
> **Q3**: A broader spectrum that can be analyzed , e,g. using examples from different classes, then examples from within classes
>
> **A3**: Thank you for the insightful suggestion to broaden the analysis spectrum. In response, based on foreground segmentation task, we conducted random sampling experiments with three distinct scenarios:
>
> (a) examples randomly sampled from the entire training dataset
>
> (b) examples randomly sampled across different classes
>
> (c) examples sampled from within classes.
>
> Our findings indicates that the alignment of example data within the same classes as the query data is crucial. This reaffirms our observation that an effective in-context example should be semantically similar to the query.
>
> |            | Different classes | Whole training data | Within classes | UnsupPR | SupPR
> |------------|-------------------|---------------------|----------------|-------|------|
> | Seg.(mIoU) | 24.02             | 24.46               | 27.45          | 33.56 | 35.56

---

> > ### Comment · Reviewer_EQqT · 2023-08-13
> > **Response to authors**
> >
> > Thank you for addressing my concerns, following the author's rebuttal I raise my score and I now support the acceptance of the paper.

---

> > > ### Author Response · Authors · 2023-08-14
> > > **Thanks for raising your score.**
> > >
> > > Thanks for raising your score! We’re very encouraged that our rebuttal addressed your concerns and appreciate your support for the paper's acceptance.

---

### Official Review · Reviewer_QAph · 2023-07-10

**Soundness:** 3 good
**Presentation:** 3 good
**Contribution:** 3 good
**Rating:** 4
**Confidence:** 4

**Summary:**

This paper investigates in-context learning for large vision models. Specifically, the authors propose to automatically retrieve prompt for vision models in two methods: (1) nearest example search with off-the-shelf model, (2) supervised prompt retrieval method.
Experimental results show that the proposed method bring non-trivial improvement, comparing to the random selection on foreground segmentation, single object detection and colorization.

**Strengths:**


- the motivation of this paper is important: investigating how vision models could benefit from in-context learning.

- the writing is clear and easy to follow.

- The experiments show improvement comparing with random selection.



**Weaknesses:**

1. Visual in-context learning emerged from large autoregressive language models. In-context learning was referred to use language models for a wide range of downstream tasks without updating the model itself. But this paper, though claims to study visual in-context learning, only studies a limited set of tasks (assign labels to pixels) with in-paining model. I wonder if the authors could provide thoughts on how to apply to a wider visual tasks, like classification, or standard segmentation tasks (semantic and instance).


2. The finding of "a good in-context example should be semantically similar to query and closer in context" seems to be a problem if the testing examples are unique, and there is no semantically similar {image, annotation} pair in the retrieval pool. This goes to the practicality of the method, and goes to the claim "potential of using prompt retrieval in vision applications".





**Questions:**

please refer to the weakness section

**Limitations:**

yes

---

> ### Author Rebuttal · Authors · 2023-08-08
>
> We thank the reviewer for the feedback and comments. We respond to the concerns below:
>
> **Q1**:Only studies a limited set of tasks (assign labels to pixels) with an inpaining model; to a wider visual tasks, like classification, or standard segmentation tasks (semantic and instance).
>
> **A1**: Thanks for your suggestion. We would like to clarify that the SupPR/UnsupPR model primarily aims to enhance the in-context learning capabilities emerged during pre-training, rather than creating new abilities. Consequently, we posit that our model could be feasibly extended to standard segmentation tasks, provided the pre-trained inpainting model demonstrates in-context learning on such tasks, albeit its current limitation in standard segmentation tasks.
>
> Additionally, our method shows potential for extension to a broader range of visual tasks. The UnsupPR/SupPR can be easily extended to other tasks. As detailed in the **General Response** section, we extend our method to image caption task and VQA task based on a vision-language model. We appreciate your feedback and encourage you to refer to that section for a comprehensive understanding.
>
> **Q2**: Seems to be a problem if the testing examples are unique
>
> **A2**: We appreciate the reviewer's point regarding unique testing examples. However, it is essential to note that our method is fundamentally designed for scenarios where there are a certain amount of testing examples, which aligns with the standard presumption of in-context learning. Also, due to the relative nature of similarity, the most semantically similar pair can always be identified within the retrieval pool. Our Figure 5 (left) illustrates that even with a retrieval set of just 0.01% of the entire dataset (about 30 image-annotation pairs), which may not include the most semantically similar pairs, supPR and UnsupPR consistently surpass random selection. This highlights the resilience and practicality of our method.

---

> > ### Author Response · Authors · 2023-08-14
> > **Follow-up**
> >
> > Dear Reviewer,
> >
> > In our rebuttal:
> >
> > 1) Regarding concerns about the limited set of tasks (assign labels to pixels) with an inpaining model:
> >
> > - Our method has demonstrated easy generalization to visual-language tasks such as image captioning and VQA.
> >
> > 2) On the point of  "Seems to be a problem if the testing examples are unique":
> >
> > - We underscore that (1) Our study follows the in-context learning paradigm, where a given set of testing examples is assumed. (2) Even without perfect semantically similar pairs in the retrieval pool, our SupPR and UnsupPR methods consistently outperform random selection, underscoring our method's practicality.
> >
> > We would love your feedback on whether our answer has solved your concern or if you have further questions.

---

> > > ### Author Response · Authors · 2023-08-17
> > > **Follow-up**
> > >
> > > Dear reviewer,
> > >
> > > In our rebuttal, we hope we have effectively clarified your confusion, and our added experiments serve to bolster the strength of our approach.
> > >
> > > Given that our rebuttal has effectively resolved concerns raised by other reviewers, we eagerly await your input on whether our response adequately addresses your apprehensions, or if you require additional clarification.

---

> > > > ### Author Response · Authors · 2023-08-19
> > > > **Follow-up**
> > > >
> > > > Dear reviewer,
> > > >
> > > > With the discussion phase approaching close, we would be greatly appreciated if you can engage in discussion timely. We hope we can address your concern in the discussion session.

---

### Official Review · Reviewer_mHCs · 2023-07-19

**Soundness:** 3 good
**Presentation:** 3 good
**Contribution:** 2 fair
**Rating:** 5
**Confidence:** 5

**Summary:**

This paper aims to address the problem of vision in-context-learning that the performance highly depends on the choice of visual in-context examples. In the paper, the authors propose automatically retrieving prompts in unsupervised and supervised ways without reaching internal weights of large vision models. Besides, this paper comprehensively studies how to select good examples for visual in-context learning and shares some valuable insights with the community on choosing good visual in-context examples.

**Strengths:**

The paper discusses several aspects that influence visual in-context learning and propose methods to choose in-context learning samples automatically to optimize visual in-context learning with inpainting method. The proposed methods both outperform random selection baseline and the supervised version requires training performs better than the unsupervised method in several tasks. Also, this paper discusses other factors that might influence visual in-context-learning including the number of examples, order of examples, and size of retrieval set, which shares some useful practical experience to the community.

**Weaknesses:**

1. This paper only discusses the previous paper of using inpainting as visual in-context learning as the visual icl frame and experiment on the dataset from that paper. It's hard to tell if experiments and conclusions conducted on a specific framework will generalize to more general "visual in-context learning" settings. Besides, although the supervised example retrieval method outperforms the unsupervised one, the additional model and training seem to be contradictory to the main advantages of ICL learning that requires no additional training.

**Questions:**

1. The paper discusses the impact of the order and number of in-context examples. While more in-context examples generally lead to better performance, is there an upper limit to this? Will adding more examples become redundant or even detrimental at some points?

2. The paper compares the proposed methods with random selection. Are there other baselines that the authors considered for comparison?



**Limitations:**

Yes

---

> ### Author Rebuttal · Authors · 2023-08-08
>
> We thank the reviewer for the feedback and comments. We respond to the concerns below:
>
> **Q1**:Whether this method can be generalized to more general "visual in-context learning" settings.
>
> **A1**:Thanks for your suggestion. Please refer to the **General Response** for the detailed explanation.
>
> **Q2**: SupPR contradicts the main advantages of ICL learning that requires no additional training
>
> **A2**: The advantages of In-context learning (ICL) comes from no additional training for the **pre-trained model** that might be only provided as the service, e.g. ChatGPT, Flamingo, PaLM2. SupPR only uses the pre-trained model as a mechanism for in-context results output, while training an independent scoring model for prompt retrieval. This approach uses downstream few-shot data effectively and do not train the pre-trained model. The goal of SupPR  is to enhance the ICL capability (such as foreground segmentation) emerged during pre-training rather than creating new capabilities.
>
> **Q3**: The upper bound of the number of in-context examples.
>
> **A3**: We appreciate the reviewer's suggestion to investigate the upper bound of the number of in-context examples. In response, we conducted an additional experiment focusing on the foreground segmentation task, where we employed SupPR as the prompt retrieval method. We increased the maximum number of in-context examples to 32. However, we would like to note that any further increase in in-context examples would result in out-of-memory errors on NVIDIA V100 GPUs. The findings, presented below, detail the performance improvement with an increase in the number of shots. The performance enhancement achieved from the latest shot is indicated within round brackets. From this table, it is evident that while there is an almost linear increase in performance from 1 shot to 16 shots, the improvement diminishes from 16 shots to 32 shots.
>
> |            | 1 shot | 4 shot        | 8 shot       | 16 shot      | 32 shot      |
> |------------|--------|---------------|--------------|--------------|--------------|
> | Seg.(mIoU) | 21.90  | 25.04 (+3.1) | 28.45 (3.4) | 31.25 (+2.8) | 32.64 (+1.4) |
>
> **Q4**: Other baselines that the authors considered for comparison
>
> **A4**: Thank you for your query on baselines. In line with Bar et al.[1], our primary baseline employs a within-class random selection of prompts matching the query image class. To broaden our comparisons, we introduced another baseline where prompts are randomly chosen from the entire training dataset. The significant performance difference between UnsupPR/SupPR and this (entire) Random baseline underscores the effectiveness of our method in real-world scenarios.
>
> |            | (entire) Random | (within-class) Random | UnsupPR | SupPR |
> |------------|-----------------|-----------------------|---------|-------|
> | Seg.(mIoU) | 24.46           | 27.45                 | 33.56   | 35.56 |
>
> [1] Bar, Amir, et al. "Visual prompting via image inpainting." Advances in Neural Information Processing Systems 35 (2022): 25005-25017.

---

> > ### Author Response · Authors · 2023-08-14
> > **Follow-up**
> >
> > Dear Reviewer,
> >
> > In our rebuttal:
> >
> > 1) Regarding concerns about the generalization of our method to broader "visual in-context learning" settings:
> >
> > - Our method has demonstrated easy generalization to visual-language tasks such as image captioning and VQA.
> >
> > 2) On the point of "additional training" potentially contradicting ICL advantages:
> >
> > - We claim that since the SupPR doesn't train the pre-trained model, it does not conflict with the benefits of ICL.
> >
> > Furthermore, we've incorporated experiments exploring:
> >
> > 1) The upper bound of in-context examples:
> >
> > - Our findings indicate an almost linear performance increase from 1-shot to 16-shots. However, the gains plateau from 16 to 32 shots.
> >
> > 2) Additional baseline comparisons:
> >
> > - We add a baseline about randomly selecting the in-context examples from the entire training dataset, which further strengthen the superiority of our proposed UnsupPR and SupPR.
> >
> > We would love your feedback on whether our answer has solved your concern or if you have further questions.

---

> > > ### Author Response · Authors · 2023-08-17
> > > **Follow-up**
> > >
> > > Dear reviewer,
> > >
> > > In our rebuttal, we hope we have effectively clarified your confusion, and our added experiments serve to bolster the strength of our approach.
> > >
> > > Given that our rebuttal has effectively resolved concerns raised by other reviewers, we eagerly await your input on whether our response adequately addresses your apprehensions, or if you require additional clarification.

---

> > ### Comment · Reviewer_mHCs · 2023-08-18
> >
> > Thank you for the aditional experiments and reply, I think the rebuttal basically address my concerns and I decide to raise my rating to borderline accept.

---

> > > ### Author Response · Authors · 2023-08-18
> > > **Thanks for raising your score**
> > >
> > > Thanks for raising your score! We’re very encouraged that our rebuttal basically addressed your concerns and appreciate your support for the paper's acceptance.

---

### Author Rebuttal · Authors · 2023-08-08

**General Response**

Dear Reviewers,

We sincerely appreciate the time and insightful comments provided by all reviewers, which have been instrumental in enhancing our paper. We are encouraged by the positive feedback regarding the motivation behind our work (QAph) and our strong performance across various tasks (mHCs, QAph, EQqT, vwKB), our analysis is useful, interesting and insightful (mHCs, vwKB).

**We would like to highlight the key contributions of our work**

(1) Our study is the first comprehensive examination of selecting effective examples for the emerging field of visual in-context learning. We uncover a critical issue: the choice of in-context examples significantly impacts performance.

(2) From a technical standpoint, we introduce a prompt retrieval framework that automates the prompt selection process, offering two straightforward implementations: an unsupervised method and a supervised method.

(3) Through extensive experimentation on three visual in-context learning tasks not encountered during pre-training (foreground segmentation, single object detection, and image colorization), we provide the community with valuable insights on identifying suitable visual in-context examples. For instance, our findings indicate that the supervised method consistently outperforms other approaches and often identifies examples that are both semantically related and/or spatially similar to a given query.

**Our methods are adaptable to broader "Visual In-Context Learning" scenarios**

Our strategies - SupPR and UnsupPR - are designed with a broad scope of applicability , capable of extending to a variety of "visual in-context learning" frameworks. We have substantiated this claim by conducting experiments on the COCO Caption dataset and the TextVQA dataset, utilizing the recently public OpenFlamingoV2-9B model [1]. OpenFlamingo, a visual-language model distinct from the inpainting model, can generate natural language description from visual-language inputs.

In these experiments:

- For the UnsupPR method, we used the CLIP ViT L-14 as the feature extractor.
- For SupPR, a prompt is defined either by an image-caption pair (COCO caption) or by image-question-answer triplets (TextVQA). A prompt that instructs a query image for the maximum/minimum CIDEr (COCO caption) or accuracy (TextVQA) is labeled as positive/negative prompt. Using these positive/negative prompts, we trained a new feature extractor following the methodologies outlined in Sec.2.2.2.

The comparative performance between random selection and our proposed SupPR and UnsupPR are presented below. The results indicates the broader applicability of our methods to more visual in-context learning task and visual model.

|                      | Random | UnsupPR | SupPR |
|----------------------|--------|---------|-------|
| COCO caption (CIDEr) | 84.2   | 87.3    | 88.9  |
| TextVQA (Acc.) | 25.4 | 30.2 | 33.4 |

Once again, we thank the reviewers for their invaluable feedback and support.

[1] Awadalla, Anas et al. “OpenFlamingo: An Open-Source Framework for Training Large Autoregressive Vision-Language Models.” (2023).

---

### Decision · Program_Chairs · 2023-09-21

**Decision:**

Accept (poster)

**Comment:**

Thank you for addressing the concerns highlighted during the review phase in detail.

From your response, it's clear that you've actively engaged with the feedback, particularly regarding the method's extendability. The majority of reviewers are content with your clarifications and recognize the novelty, robustness, and relevance of your method within the "Visual In-Context Learning" context. Your focus on the importance of in-context example selection offers pivotal guidance for subsequent research in this area.

We're pleased to accept your paper, confident in its potential to significantly influence the visual in-context learning domain.

Area Chair, NeurIPS.